



# Water isotopes – climate relationships for the mid-Holocene and pre-industrial period simulated with an isotope-enabled version of MPI-ESM

Alexandre Cauquoin[1], Martin Werner[1], and Gerrit Lohmann[1]

[1]Alfred Wegener Institute, Helmholtz Centre for Polar and Marine Sciences, Bremerhaven, Germany

**Correspondence:** Alexandre Cauquoin (alexandre.cauquoin@awi.de)

**Abstract.** We present here the first results, for the pre-industrial and mid-Holocene climatological periods, of the newly developed isotope-enhanced version of the fully coupled Earth system model MPI-ESM, called hereafter MPI-ESM-wiso. The water stable isotopes $H_2^{16}O$, $H_2^{18}O$ and HDO have been implemented into all components of the coupled model setup: the atmosphere model ECHAM6, the land/soil vegetation model JSBACH, and the ocean/sea ice model MPIOM. The exchanges of the

related isotope masses between the atmosphere and the ocean are made via the coupler OASIS3. The mid-Holocene, one of the PMIP4-CMIP6 entry cards to evaluate the performance of the latest generation of fully-coupled General Circulation Models, provides the opportunity to evaluate the model response to changes in the seasonal and latitudinal distribution of insolation induced by different orbital forcing conditions. The results of our equilibrium simulations allow to evaluate the performance of the isotopic model in simulating the spatial and temporal variations of water isotopes in the different compartments of the

hydrological system for warm climates. It represents a first necessary step before simulating other climatological interglacial periods or transient Holocene experiment. For pre-industrial climate, MPI-ESM-wiso reproduces very well the observed spatial distribution of isotopic content in precipitation, in link with the spatial variations in temperature and precipitation rate. We find also a good model-data agreement with the observed distribution of isotopic composition in surface seawater, but a bias with too depleted surface seawater is present in the Arctic Ocean. All these results are improved compared to the previous model

version ECHAM5/MPIOM. The spatial relationships of water isotopic composition with temperature, precipitation rate and salinity are consistent with observational data. For the pre-industrial climate, the interannual relationships of water isotopes with temperature and salinity are globally lower than the spatial ones, consistent with previous studies. Simulated results under mid-Holocene conditions are in fair agreement with the isotopic measurements from ice cores and continental speleothems. MPI-ESM-wiso simulates a depletion in isotopic composition of precipitation from North Africa to the Tibetan plateau via

India due to the enhanced monsoons during mid-Holocene. Over Greenland, our simulation indicates enriched isotopic composition of precipitation over Greenland in link with higher summer temperature and reduction in sea ice, shown by positive isotope-temperature gradient. For the Antarctic continent, the model simulates depleted isotopic values over the East Antarctic plateau, in link with the lower temperatures during the mid-Holocene period, while similar or higher isotopic values are modeled over the rest of the continent. While variations of isotopic contents in precipitation over West Antarctica between

mid-Holocene and pre-industrial periods are partly controlled by changes in temperature, the transport of relatively enriched





water vapor near the coast to the western ice core sites could play a role in the final isotopic composition. The reconstruction of past salinity through isotopic content in sea surface waters can be complicated for regions with strong ocean dynamics, variations in sea ice regimes or significant changes in freshwater budget, giving an extremely variable relationship between isotopic content and salinity of ocean surface waters over small spatial scales. These complicating factors demonstrate the complexity

in interpreting water isotopes as past climate signals of warm periods like the mid-Holocene.

## 1 Introduction

The hydrogen and oxygen atoms that compose the water molecule have several natural stable isotopes. This results in several forms of the water molecule called water stable isotopologues (hereafter designated by the term "water isotopes"), the most common being $H_2^{16}O$, $H_2^{18}O$ and HDO. These water isotopes, expressed hereafter in the usual $\delta$-notation (as $\delta^{18}O$ and $\delta D$

with respect to the Vienna Standard Mean Ocean Water V-SMOW, if not stated otherwise), are integrated tracers of climatic processes occurring in diverse parts of the hydrological cycle (Craig and Gordon, 1965; Dansgaard, 1964). Because of their differences in mass and symmetry, an isotopic fractionation happens at each phase change depending on environmental conditions. As a consequence, the water isotopes have been successfully used during the last decades to study past climate changes and to describe the present-day water cycle through their measurements in various natural archives. Many of these studies

are based on a modern analogue approach, i.e. by assuming that the modern spatial relationship between water isotopes and surface temperatures, precipitation amount or salinity provides a calibration that can be used for different past climates. In addition to be consistent with the observed close relationships between water isotopic time series and temperature or precipitation amount variations, this hypothesis can be validated by a Rayleigh distillation model representing the evolution of the remaining water vapor in a cloud (i.e. loss of heavier isotopes during condensation and precipitation events) as it is transported

from moisture source region to high latitudes (Ciais and Jouzel, 1994). For example, the isotopic signal measured in polar ice cores enabled at a first order the reconstruction of past temperature variations at high resolution (Jouzel, 2013, and references therein), allowing the description of past climate changes over several glacial-interglacial periods (Jouzel et al., 2007; NEEM Community Members, 2013). In the (sub-)tropical areas, the $\delta^{18}O$ in the calcite of speleothems is interpreted in terms of past monsoon dynamics (i.e. linked to the quantity of precipitation, called "amount effect") (Wang et al., 2001, 2008). Analogously

to the continental speleothems, the $\delta^{18}O$ conserved in the carbonates of foraminifers or corals can be measured. It is controlled by the $^{18}O$ isotopic composition of ocean water and the temperature at the calcite formation. Such records from marine sediment cores are essential to deduce mean sea-level changes which are linked to the global ice volume during different climates (Shackleton, 1967). Moreover, the local variations in the $\delta^{18}O$ of ocean water tend to be dependent on changes in freshwater budget and ocean circulation, and so provide information about salinity changes. Finally, the combination of $\delta D$ and $\delta^{18}O$

measured in a same sample gives access to the second-order parameter Deuterium-excess (dex), defined as dex $= \delta D - 8 \times \delta^{18}O$ (Dansgaard, 1964). Deuterium-excess changes are often interpreted as a source region effect, i.e. dex is related to the humidity and temperature conditions at the evaporative source regions (Merlivat and Jouzel, 1979).



However, the quantitative translation of past isotope signals recorded in natural archives to climate variables and their interpretation remain challenging because of the numerous and complex processes involved: changes in evaporation conditions and moisture sources, in atmospheric transport pathways, or in the seasonality of the precipitation. For example, using the spatial relationship between the $\delta^{18}$O in Greenland ice core records and surface temperature to evaluate the local temperature variations during the last deglaciation leads to a large uncertainty of a factor 2 (Jouzel, 1999; Buizert et al., 2014). This has been attributed to changes in air mass origins (Werner et al., 2001), precipitation seasonality (Krinner et al., 1997; Krinner and Werner, 2003) or to a dampening of isotopic changes by ocean evaporation (Lee et al., 2008). In East Antarctica, it has been suggested that the relationship between temperature and the isotopic signature for warmer interglacial periods can vary among ice core sites, with an error on the temperature reconstruction that can reach up to 100 % (Sime et al., 2009; Cauquoin et al., 2015). At lower latitudes, the interpretation of water isotope records is even more complex because of the convective processes (Risi et al., 2008) and of the importance of the precipitation intensity that affect the isotopic composition of these records (Vimeux et al., 2005). In the oceans, the quantitative reconstructions of past salinity variability based on its spatial relationship with $\delta^{18}$O in ocean water may have very large errors and uncertainties, too (LeGrande and Schmidt, 2011).

One way to improve our understanding of the mechanisms controlling the water isotopes distribution in link with the variations of climate is to use General Circulation Models (GCMs) with explicit diagnostics of water stable isotopes. These complex models consider the numerous physical processes that influence the isotopic composition of the different water bodies in the Earth's climate system. Since the pioneering work of Joussaume et al. (1984), several isotope-enabled GCMs have been built both for the atmosphere (Jouzel et al., 1987; Hoffmann et al., 1998; Mathieu et al., 2002; Noone and Simmonds, 2002; Schmidt et al., 2005; Lee et al., 2007; Yoshimura et al., 2008; Risi et al., 2010b; Werner et al., 2011; Kurita et al., 2011; Nusbaumer et al., 2017) and the ocean (Schmidt, 1998; Paul et al., 1999; Delaygue et al., 2000; Xu et al., 2012; Liu et al., 2014). These models are extremely powerful because they make it possible to perform direct comparisons, at different time periods, with environmental proxy records and to reduce the uncertainties resulting from the interpretation of these records in terms of climate signals in model-data comparisons. They have been used for a considerable range of applications: e.g., analyses of mixing processes within rain events (Risi et al., 2010a), an estimation of the changes in temperature and ice sheet height in Antarctica during the last glacial period (Werner et al., 2018), or a study of the link between oceanic water isotopic content and salinity, which is of crucial interest in paleoceanography (Delaygue et al., 2000).

When simulating different climates or evolving climate conditions, it is essential to describe in a coherent way the numerous links and feedbacks between the different natural reservoirs (atmosphere, land/vegetation, ocean) and to minimize the prescription of unknown boundary conditions (e.g., sea surface temperatures). For paleoclimate isotope applications, it means that it is necessary to simulate the water isotopes in a full hydrological cycle system, not only in the atmosphere or in the ocean components. With the gain in performance of supercomputers, it is now possible to model the water isotopes in fully coupled atmosphere-ocean GCMs. In the past decade, such models have been used to examine the internal variability and the forced response to orbital and greenhouse gas forcing for modern and mid-Holocene (6 ka) climates (GISS Model E: Schmidt et al. (2007)), and to study the isotopic signature of El Ninõ-Southern Oscillation in link with the tropical amount effect (HadCM3: Tindall et al. (2009)). More recently, the isotopic-enabled model HadCM3 has been used to reconstruct past paleosalinity from





modeled $\delta^{18}O$ in ocean water during the modern period, the last glacial maximum (LGM, 21 ka) and the last interglacial optimum (LIG, 130 to 115 ka) (Holloway et al., 2016) and to investigate the magnitude of Antarctic warming in response to Northern Hemisphere meltwater input at 128 ka (Holloway et al., 2018). With the same model, Sime et al. (2019) confirm the primary importance of sea ice as a control on southern Greenland ice core $\delta^{18}O$ during the Dansgaard-Oescher events. Using

the ECHAM5/MPIOM model, Werner et al. (2016) have examined the changes in $\delta^{18}O$ and dex between the LGM and the modern period. This same model has been exploited to examine the $\delta^{18}O$–temperature temporal relationship between the LIG and the modern period (Gierz et al., 2017).

     The mid-Holocene (6k) is one of the PMIP4-CMIP6 (Paleoclimate Modeling Intercomparison Project – Coupled Model Intercomparison Project) past climates to evaluate the performance of the coupled GCMs (Kageyama et al., 2018). The mid-

Holocene climate provides the opportunity to evaluate the model response to changes in the seasonal and latitudinal distribution of insolation induced by different orbital parameters. Due to a larger obliquity 6000 years ago and changes in the precession (Berger, 1978), the amplitude of the insolation seasonal changes is amplified in the Northern Hemisphere according to the increase in boreal summer insolation and the decrease in boreal winter insolation. This is the opposite for the Southern Hemisphere. So, the mid-Holocene is characterized by an enhanced seasonal contrast in the Northern Hemisphere with warmer

summers and a significant enhancement of the monsoons in this part of the Earth. Even if the forcing mechanisms are not linked to anthropogenic actions, a better quantification of the contributions of the orbital forcing variations and their related feedbacks on large-scale climate variations like the amplification in seasonal temperature changes and the related responses of the hydrological cycle and of the oceanic circulation, is still an important issue that is relevant for evaluating future climate projections. A good way to progress on these questions is to investigate the variability of the isotope-to-climate gradients (spatial

and temporal) for warm climatic periods under different orbital forcing conditions like PI and 6k.

     In this paper, we present the first results of a new isotope-enhanced version of the fully coupled model MPI-ESM (Giorgetta et al., 2013; Mauritsen et al., 2019), called hereafter MPI-ESM-wiso. It follows the efforts of Werner et al. (2016) who developed the previous model version. The better performance of the presently available supercomputers combined with an optimization of computational cost of the model allow us to run MPI-ESM-wiso with a finer spatial horizontal resolution

compared to the others isotope-enabled fully coupled models (e.g., the horizontal resolution is two times better than for the ECHAM5/MPIOM model setup used by Werner et al. (2016)). Our study focuses on isotope changes and isotope-climate relationships for the mid-Holocene and pre-industrial period. The outline of the paper is as follow. In Section 2, we briefly describe the model components, the implementation of water isotopes and the dataset used for model evaluation. In Section 3, we evaluate MPI-ESM-wiso simulation results. We present the simulated spatial variations of water isotopes in the atmospheric

and oceanic compartments for both pre-industrial and mid-Holocene periods and compare them with available observations. We also analyze their spatial relationships with climate variables like near-air surface temperature and salinity. In Section 4, the temporal relationships between water isotopes and climate variables are analyzed during and between the mid-Holocene and pre-industrial periods. We conclude the article with a summary of our findings and some perspective remarks in Section 5.





## 2 Model Simulations and Data Sets

### 2.1 MPI-ESM-wiso

For this study, we have implemented the water stable isotopes in the Earth system model MPI-ESM (Giorgetta et al., 2013; Mauritsen et al., 2019), version 1.2.01p1. It consists of the components ECHAM6 (Stevens et al., 2013)(Stevens et al., 2013) for

the atmosphere and MPIOM (Jungclaus et al., 2013) for the ocean, as well as JSBACH (Reick et al., 2013; Schneck et al., 2013) for the land and vegetation and HAMOCC (Ilyina et al., 2013) for the marine biogeochemistry. The coupling of atmosphere and land processes on the one hand and physical ocean and biogeochemistry on the other hand is done by the OASIS3 coupler (Valcke, 2013). MPI-ESM has been used for a wide range of CMIP5 experiments and will participate in CMIP6/PMIP4 with different model configurations (i.e. resolutions) and experiments (Eyring et al., 2016; Kageyama et al., 2018).

To explicitly simulate both $H_2^{18}O$ and HDO within the different parts of the hydrological cycle, MPI-ESM has been equipped with water isotopes diagnostics in each of its components in the same way as in the previous model version (ECHAM5, JSBACH, MPIOM) (Werner et al., 2016). Here, we give a brief summary of key model components, including their differences with the previous model setup, and isotope implementation within them. As the physical and dynamical processes in the water cycle are only involved in the ECHAM6, JSBACH and MPIOM components, we do not consider HAMOCC

in the following descriptions.

The atmospheric general circulation model ECHAM6 has been developed on the basis of ECHAM5 (Roeckner et al., 2003). The detailed description of the model is given by Stevens et al. (2013). ECHAM6 consists of a dry spectral-transform dynamical core, a transport model for scalar quantities other than temperature and surface pressure, a suite of physical parameterizations for the representation of diabatic processes, as well as boundary data sets for externalized parameters (trace gas and aerosol

distributions, land-surface properties, etc.) (Stevens et al., 2013). The most important differences between ECHAM5 and ECHAM6 concern the radiation schemes with an improved representation of radiative transfer in the solar part of the spectrum, the computation of surface albedo, a new aerosol climatology, and an improved representation of the middle atmosphere. Moreover, minor changes have been made in the representation of convective processes, and through the choice of a slightly different vertical discretization within the troposphere. As in ECHAM5, the water cycle in ECHAM6 contains formulations

for evapotranspiration of terrestrial water, evaporation of ocean water, and the formation of large-scale and convective clouds. Within the atmosphere's advection scheme, vapor, liquid, and frozen water are transported independently. Water stable isotopes $H_2^{16}O$, $H_2^{18}O$ and HDO have been explicitly implemented into the hydrological cycle of ECHAM6 in an analogous manner to the previous model release ECHAM5 (Werner et al., 2011). The water isotopes are implemented parallel to the "normal" water cycle: the isotopes are described identically as the "normal" water as long as no phase transitions are concerned. Additional

fractionation processes are defined for the water isotope variables whenever a phase change of the "normal" water occurs. The equilibrium fractionation coefficients between vapor and liquid/ice water are calculated from Merlivat and Nief (1967) and Majoube (1971a, b). The kinetic (i.e. non-equilibrium) effects during evaporation from ocean sea surface and snow formation follow the formulations of Merlivat and Jouzel (1979) and Jouzel and Merlivat (1984) respectively. For the latter, we use the



same supersaturation function as Werner et al. (2011). In the coupled set-up, ECHAM6 provides the required freshwater flux (net precipitation P−E) and its isotopic composition for all ocean grid cells to the MPIOM ocean model.

The land-surface model JSBACH (Reick et al., 2013; Schneck et al., 2013) calculates the boundary conditions for ECHAM6 over terrestrial areas. It simulates water, energy, and carbon related processes including interactive and dynamic vegetation,
that is controlled by the processes of natural growing and mortality, as well as disturbance mortality (e.g. wind, fire) (Brovkin et al., 2009; Reick et al., 2013). The physical processes partitioning water masses on the land surfaces comprise the separation of rainfall and snowmelt into surface runoff and infiltration and the calculation of lateral drainage. Contrary to the previous release of JSBACH, the soil hydrology is now simulated similarly to the soil temperatures within 5 soil layers (Hagemann and Stacke, 2015) with increasing thickness (0.065, 0.254, 0.913, 2.902, and 5.7 m), the lower boundary being at almost 10 m
depth. The isotopic processes are represented in the same way as described in Werner et al. (2016), i.e. the water isotopes are passive tracers in the JSBACH model. No fractionation of the isotopes is assumed during most physical processes partitioning water masses on the land surface: the surface runoff has the isotopic composition of the rainfall and snow melt that reach the soil surface, and drainage has the isotopic composition of soil layer water (Haese et al., 2013). The water that percolates by gravitational drainage from one soil layer $z$ to the layer below $z+1$ has the isotopic composition of moisture content in the
layer $z$. The transport of $H_2^{16}O$, $H_2^{18}O$ and HDO between the different layers via the vertical diffusion is treated in the same way as for the standard water. For evapotranspiration, fractionation of isotopes might occur during evaporation of water from bare soils (i.e. from the surface soil layer). However, the strength of this fractionation remains an open question. In accordance with the results of Haese et al. (2013) and as explained by Werner et al. (2016), we assume in this study that we can ignore any possible fractionation during evapotranspiration processes from terrestrial areas, as our analyses will focus primarily on
the isotopic composition of precipitation.

As a part of the coupled model MPI-ESM, the Hydrological Discharge (HD) model (Hagemann and Gates, 2003) globally simulates the lateral freshwater fluxes at the land surface that go to the ocean at a daily time step. Modelled water discharge is calculated with respect to the topography gradient between grid boxes, the slope within a grid box, the grid box length, the lake area and the wetland fraction of a particular grid box. For the simulated total river runoff, it is assumed that the global
water cycle is closed, i.e. that all net precipitation (P−E) over terrestrial areas is transported to the ocean. As MPI-ESM does not include a dynamic ice sheet model, precipitation amounts falling on glaciers are instantaneously put as runoff into the nearest ocean grid cell to close the global water budget. The HD model computes the discharge at 0.5° horizontal resolution. The model input fields for runoff and drainage resulting from the ECHAM6 resolution (such as T63 in this study) are therefore interpolated to the same 0.5° grid. Water stable isotopes are incorporated as passive tracers within the HD scheme.
The ocean-component, MPIOM, has remained unchanged, except for the adaptations to high-resolution grids (Jungclaus et al., 2013). MPIOM is a free-surface ocean general circulation model formulated on an Arakawa-C grid in the horizontal and a z-grid in the vertical. It contains subgrid-scale parameterizations for convection, vertical and isopycnal diffusivity, horizontal and vertical viscosity, as well as for the bottom boundary layer flow across steep topography. MPIOM includes a sea-ice model formulated using the viscous-plastic rheology of Hibler (1979). Sea-ice thermodynamics relate changes in sea-ice thickness
to a balance of radiant and turbulent atmospheric fluxes, and oceanic heat fluxes. The effect of snow accumulation on sea





ice is included, along with snow-ice transformation. As in the previous model version (Xu et al., 2012; Werner et al., 2016), $H_2^{16}O$, $H_2^{18}O$ and HDO are treated as conservative passive tracers within MPIOM. The isotopic variations occurring in this component depend on oceanic advection and mixing of different water masses, on the isotopic composition of freshwater fluxes entering in the ocean (P−E and runoff discharge), and on the temperature-dependent isotope fractionation during evaporation.

The isotopic composition of sea-ice, formed from liquid waters, is also calculated by a liquid to ice equilibrium fractionation factor of 1.003, which is the average from various estimates (Craig and Gordon, 1965; Lehmann and Siegenthaler, 1991; Macdonald et al., 1995; Majoube, 1971a). Due to the very low rate of isotopic diffusion in sea ice, we assume no fractionation during sea ice melting. In a coupled setup, MPIOM provides the isotopic composition of ocean surface water and sea-ice as boundary conditions to the ECHAM6 atmosphere model.

The coupling procedure between the atmosphere and the ocean in MPI-ESM, via the OASIS3 coupler (Valcke, 2013), has remained unchanged compared to the ECHAM5/MPIOM model setup. Mass, energy, and momentum fluxes, as well as the related isotope masses of $H_2^{16}O$, $H_2^{18}O$ and HDO, are exchanged between the atmosphere and ocean once per day.

## 2.2   Model set-up and experiments

For this study, we have used the MPI-ESM-LR configuration (LR for Low Resolution). The atmospheric component ECHAM6
was run at an approximately 1.875° horizontal resolution with 47 vertical pressure levels extending to 0.01 hPa (T63L47), while the previous T31L19 grid of ECHAM5 used by Werner et al. (2016) had a 3.75° horizontal resolution and the 19 vertical levels extended to 10 hPa. The same horizontal resolution is applied for the JSBACH land surface scheme. For the ocean component MPIOM, a bipolar grid with 1.5° horizontal resolution (near the equator) and 40 z-levels has been used (GR15L40). The poles of the ocean model are moved to Greenland and to the coast of the Weddell Sea by a conformal
mapping of the geographical grid. Again, the horizontal resolution is finer than the 3° resolution (GR30L40) used in Werner et al. (2016).

Two different experiments were performed, one for the pre-industrial period (PI), corresponding to the climate conditions at 1850 AD, and one for the mid-Holocene 6000 years ago (6k). For the pre-industrial climate, MPI-ESM has been continued from a standard PI simulation, i.e. without isotopes included, which has been run over 1000 years (C. Stepanek, personal
communication) using identical PI boundary conditions. In an analogous way as Werner et al. (2016), water isotope values in the atmosphere were initialized with constant values: $\delta^{18}O = -10$ ‰ and $\delta D = -80$ ‰. For the isotope distribution within MPIOM-wiso, we have decided to start with constant concentration values of the passive tracers $H_2^{16}O$, $H_2^{18}O$ and HDO in such a way that the respective $\delta^{18}O$ and $\delta D$ in ocean are at 0 ‰ each (Baertschi, 1976; de Wit et al., 1980). The fully coupled MPI-ESM-wiso with isotope diagnostics was then run under PI conditions according to the PMIP4 protocol (orbital forcing,
greenhouse gas concentrations, ocean bathymetry, land surface and ice sheet topography) for 2500 years. The 6k simulation is as the PI one, but with the mid-Holocene orbital and radiative active trace gas forcing according to the PMIP4 experimental design (table 1 of Otto-Bliesner et al. (2017)). Again, our isotopic simulation for 6k has been continued from a 1000-years long mid-Holocene simulation without isotopes (C. Stepanek, personal communication). The water isotope values have been initialized in the exact same way as for the PI simulation and MPI-ESM-wiso was then run for additional 2500 years.



At the end of the simulations, the global mean 2000m-deep salinity changes by less than 0.002 practical salinity unit (psu) over 100 years, and the globally averaged $\delta^{18}O$ signature at 2000m depth changes by less than 0.002 ‰/100 years. Thus, we consider both simulations as quasi-equilibrated and use the last 150 model years for our analyses. If not stated otherwise, all $\delta$ values of meteoric waters are calculated as precipitation-weighted mean with respect to the V-SMOW scale. The $\delta$ values of

ocean values are calculated as arithmetic averages, also with respect to the V-SMOW scale.

## 2.3 Observational Data

To evaluate our model, we used different datasets including isotopic measurements in precipitation, ocean water, ice cores and continental speleothems. We give a brief summary below.

For the modern isotopic content of precipitation, we use the Global Network of Isotopes in Precipitation (GNIP) database,

available through the IAEA, whose measurements have begun in the early 1960s. While several stations were monitored continuously for isotopic content of precipitation throughout several decades (e.g., Vienna station), other GNIP stations stopped their operation after a shorter period. This is why we use in this study a subset of 70 stations from this database, where surface temperature, precipitation rate, $\delta^{18}O$ and $\delta D$ have been measured for at least 5 calendar years within the period 1961 to 2007.

To evaluate the modeled PI isotope values in the ocean, we use the Goddard Institute for Space Studies (GISS) global

seawater oxygen-18 database (Schmidt et al., 1999), which is a collection of over 26 000 seawater $^{18}O$ values. We are using only the values with no applied corrections (see details in Schmidt et al. (1999). As the GISS $\delta^{18}O$ in ocean water ($\delta^{18}O_{oce}$) values do not represent annual mean values but measurements at an arbitrary day of the year, we compare them to the simulated long-term mean monthly value of the sampling month, when it is specified in the GISS database. We focus only on the near-surface $\delta^{18}O_{oce}$ values between 0m and 10m depth in this study.

Since the pioneering work of Dansgaard et al. (1969), Lorius et al. (1979) and others, the analysis of the isotopic composition of polar ice cores provided a lot of information about the climate of the past. We use here 5 Greenland and 10 Antarctic ice cores, selected from the comprehensive compilations of Sundqvist et al. (2014) and WAIS Divide Project Members (2013), to compare the measured isotopic values for the pre-industrial climate with our model results. When available, we also report the 6k−PI differences in $\delta^{18}O$. We take for PI the values averaged over the last 200 years and for 6k the average in the $6 \pm 0.5$ ka period. The ice core data used in this study are summarized in the Table 1. We also add to this ice core dataset the 6k−PI $\delta^{18}O$

anomalies measured from 4 (sub-)tropical ice cores (Huascaran, Sajama, Illimani and Guliaa ice cores), which are reported in the Table 3 of Risi et al. (2010b).

Furthermore, we also use the SISAL dataset (SISALv1b: Atsawawaranunt et al. (2019)), updated recently by Comas-Bru et al. (under review). It provides isotope records, including $\delta^{18}O$, from 455 speleothems from 211 cave sites. For our study, we

have followed the recommendation of Comas-Bru et al. (under review) by selecting 30 speleothem sites (33 cores) where averaged $\delta^{18}O$ values are available for both mid-Holocene (defined as the period $6000 \pm 500$ years BP) and pre-industrial periods (defined as the interval 1850-1990 CE). This restriction in the selected PI base period in comparison to Werner et al. (2016), who selected the last 1000 years, allows to reduce the uncertainties without substantially decreasing the available number of mean $\delta^{18}O$ speleothem values for both periods. Concerning the possible biases in the model-data comparison, a seasonal bias





**Table 1.** Selected ice core records and their geographical coordinates, reported PI values of $\delta^{18}O$ and dex, and changes in $\delta^{18}O$ and dex between 6k and PI. All values are given with respect to the V-SMOW scale.

| Site | Longitude | Latitude | $\delta^{18}O_{PI}$ (‰) | $dex_{PI}$ (‰) | $\Delta_{6k-PI}\delta^{18}O$ (‰) | $\Delta_{6k-PI}dex$ (‰) |
|---|---|---|---|---|---|---|
| Vostok[a,b] | 106.87 | −78.47 | −56.8 | 15.5 | −0.2 | − |
| Dome F[c] | 39.70 | −77.32 | −54.6 | 14.4 | 0.2 | − |
| Dome B[a] | 94.92 | −77.08 | −55 | 13.5 | − | − |
| EDC[d,e] | 123.35 | −75.10 | −50.4 | 8.6 | −0.3 | 0.7 |
| EDML[b,d] | 0.07 | −75.00 | −44.8 | 4.7 | 0.2 | − |
| Taylor Dome[f] | 158.72 | −77.80 | −40.5 | 4.9 | 1.5 | − |
| Talos[g] | 159.18 | −72.82 | −36.1 | 3.9 | −0.6 | − |
| Byrd[h] | −119.52 | −80.02 | −32.9 | 4.5 | −1.4 | − |
| Siple Dome[b] | −148.82 | −81.67 | −26.9 | 2.9 | −2.1 | − |
| WDC[b] | −112.14 | −79.46 | −34 | − | 0.5 | − |
| GRIP[a,j] | −37.63 | 72.58 | −35.3 | 9.5 | 0.4 | −0.2 |
| NGRIP[a,k] | −42.32 | 75.10 | −35.2 | 10.5 | 0.5 | −0.5 |
| Camp Century[i] | −61.13 | 77.17 | −29.3 | − | 0.8 | − |
| Dye3[j] | −43.81 | 65.18 | −27.7 | − | 0 | − |
| Renland[i] | −25.00 | 72.00 | −27.4 | − | 1 | − |

References: [a] reported in Risi et al. (2010b), [b] WAIS Divide Project Members (2013), [c] Kawamura et al. (2007), [d] Stenni et al. (2010), [e] Landais et al. (2015), [f] Steig et al. (2000), [g] Stenni et al. (2011), [h] Blunier and Brook (2001), [i] Vinther et al. (2009), [j] Vinther et al. (2006), [k] North Greenland Ice Core Project members (2004).

can appear in the isotopic composition of drip water archived in a speleothem record due to the re-evaporation of the precipitated water (Wackerbarth et al., 2010). An additional fractionation between the drip water and the formed calcite can also be observed for many speleothems (Dreybrodt and Scholz, 2011). The $\delta^{18}O$ in speleothem calcites ($\delta^{18}O_c$) is expressed with respect to the Pee Dee Belemnite (PDB) standard. To directly compare these data with our model results ($\delta^{18}O$ in precipitation:

5   $\delta^{18}O_p$), we first need to convert the $\delta^{18}O$ values in calcite between the PDB and the SMOW scale (Coplen et al., 1983):

$$\delta^{18}O_{c(PDB)} = 0.97002 \times \delta^{18}O_{c(SMOW)} - 29.98 \tag{1}$$

and then to apply a formula linking $\delta^{18}O$ in water ($\delta^{18}O_{water(SMOW)}$) and $\delta^{18}O$ in speleothem calcite (Tremaine et al., 2011):

$$\delta^{18}O_{water(SMOW)} = \delta^{18}O_{c(SMOW)} - \left( \frac{16.1 \times 1000}{T} - 24.6 \right) \tag{2}$$

with $T$ being the temperature, in Kelvin, during calcite formation. To convert the speleothem PI values of $\delta^{18}O_c$ in calcite to

10   $\delta^{18}O_p$ in precipitation, we have determined the required site temperatures by interpolating annual mean ERA40 soil temperatures (layer no. 1, mean of the period 1961–1990) to the different speleothem sites. For the direct model-data comparison of





the 6k$-$PI $\delta^{18}$O changes, we use both the simulated 6k$-$PI temperature and $\delta^{18}$O$_\mathrm{p}$ changes to calculate the modelled change in $\delta^{18}$O$_\mathrm{c}$ in calcite.

## 3   Results of the model-data comparison

### 3.1   Evaluation of MPI-ESM-wiso under PI conditions

#### 3.1.1   Water isotopes in precipitation

Fig. 1a shows the global distribution of the simulated annual mean $\delta^{18}$O$_\mathrm{p}$ values in precipitation. The main well-known patterns of the global $\delta^{18}$O$_\mathrm{p}$ distribution can be found in the model. They are very similar to those already observed with ECHAM5/MPIOM (Werner et al., 2016) and in agreement with the present-day observations (circles: GNIP, squares: ice cores, triangles: speleothems). Typically, enhanced depletion of $\delta^{18}$O$_\mathrm{p}$ with decreasing temperature (temperature effect) and

increased altitude (altitude effect) is well simulated by MPI-ESM-wiso. The lowest simulated values of $\delta^{18}$O$_\mathrm{p}$ occur over the polar regions, with the most depleted value over East Antarctica ($-54.5$ ‰). Depletion of $\delta^{18}$O$_\mathrm{p}$ is also observed going inland (continental effect) and with increased precipitation intensity over the low latitudes (precipitation amount effect).

In Fig. 1b, we compare our modeled $\delta^{18}$O$_\mathrm{p}$ with the observational dataset described in Section 2.3. The speleothem PI values of $\delta^{18}$O$_\mathrm{c}$ in calcite are converted to $\delta^{18}$O$_\mathrm{p}$ in precipitation by using the formulae (1) and (2). The modelled $\delta^{18}$O$_\mathrm{p}$ are in very

good agreement with the observations with a linear regression gradient of 0.87 (1.0 being the perfect fit) and a root-mean squared error (RMSE) of 2.3 ‰. This represents an improvement compared to the modeled results from ECHAM5/MPIOM (RMSE of 3 ‰, Werner et al. (2016)). The modelled global $\delta^{18}$O$_\mathrm{p}$–temperature relationship (for temperature below 20 °C, Fig. 1c) is also improved with a gradient 0.63 ‰.°C$^{-1}$ ($r^2 = 0.97$), very close of the observed one (0.66 ‰.°C$^{-1}$, $r^2 = 0.95$). This improvement, compared to the results from Werner et al. (2016), is mainly due to a better model-data agreement for the very

low temperatures over the poles, which constitute an extreme test for isotope-enabled GCMs. This is confirmed by the good agreement of our modeled $\delta^{18}$O$_\mathrm{p}$–temperature spatial gradient over Antarctica (0.71 ‰.°C$^{-1}$, $r^2 = 0.97$) with the gradient of 0.8 ‰.°C$^{-1}$ deduced from the Antarctic isotopic observations compiled by Masson-Delmotte et al. (2008). However, even if the warm bias for the coldest temperatures over Antarctica is reduced, the modeled $\delta^{18}$O$_\mathrm{p}$ values are still too enriched at these locations (Fig. 1b). Concerning the $\delta^{18}$O$_\mathrm{p}$–precipitation spatial gradient, we calculate observed and modeled values of

$-0.47$ and $-0.36$ ‰.mm$^{-1}$.day, respectively, for the 9 low-latitude GNIP stations with an annual mean temperature equal or above 20 °C. These results have to be taken with caution because of the few available tropical GNIP station records. The rather large standard errors of the gradients, estimated by using the variance-covariance matrix between the regression coefficients, illustrate well this point (0.165 and 0.145 ‰.mm$^{-1}$.day for GNIP and MPI-ESM-wiso results, respectively).

#### 3.1.2   Water isotopes in ocean surface waters

Fig. 2a shows the global distribution of modelled annual mean $\delta^{18}$O$_\mathrm{oce}$ in ocean surface water (mean between 0 and 10m depth) and the comparison with the observations from the GISS database (colored dots). As expected from the data, we observe higher

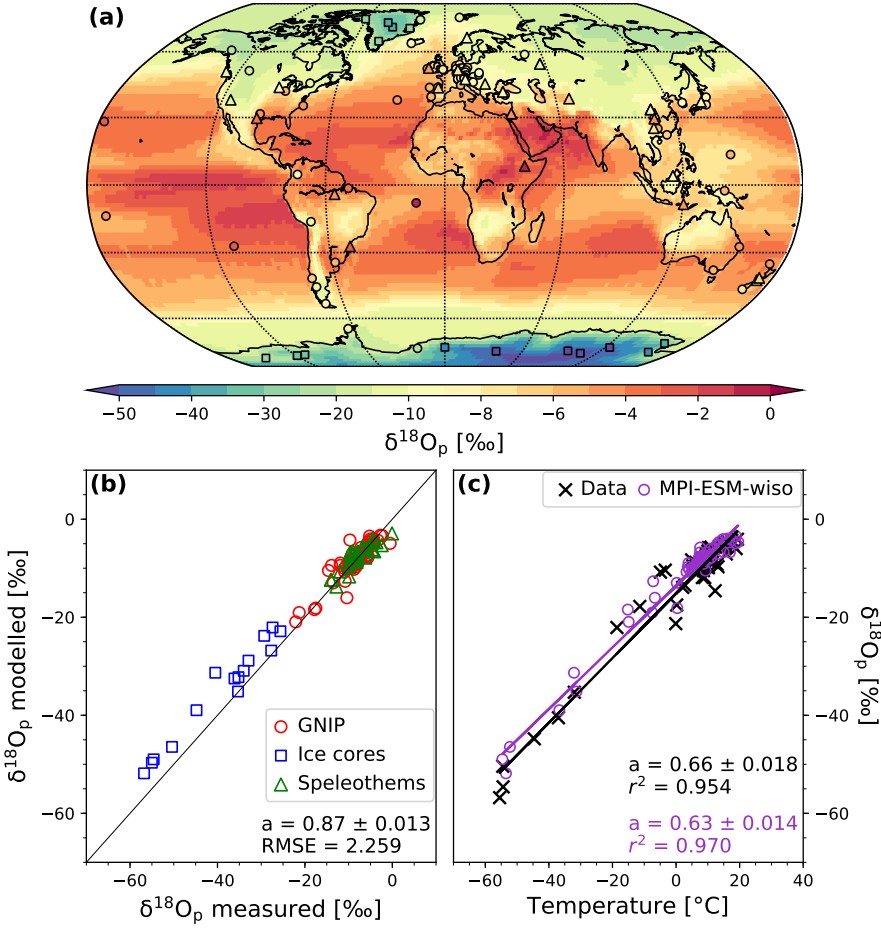

**Figure 1.** (a) Global distribution of simulated (background pattern) and observed (colored markers, see text for details) annual mean $\delta^{18}O_p$ values in precipitation under PI conditions. The data consist of 70 GNIP stations (circles), 15 ice core records (squares, Table 1) and 33 speleothem records (triangles). (b) Modelled vs. observed annual mean $\delta^{18}O_p$ at the different GNIP, speleothem, and ice core sites. (c) Observed (black crosses) and modelled (purple circles) spatial $\delta^{18}O_p$–$T$ relationship for the sites where the observed annual mean temperatures are below +20°C. The linear fits for the observed and modelled values are drawn as black and red lines respectively. For both (b) and (c), the gradients of the linear regression fits are given in the legends.

modeled $\delta^{18}O_{oce}$ values in the low to mid-latitude oceanic areas, which range between $-0.2$ ‰ in the Malaysian area and $1.1$ ‰ in the Bermuda area. The higher values in the Atlantic Ocean can be explained by a net freshwater export of Atlantic Ocean water, which is transported westwards to the Pacific (Broecker et al., 1990; Lohmann, 2003; Zaucker and Broecker, 1992). Other highly enriched $\delta^{18}O_{oce}$ values can be found in more localized areas like the Mediterranean Sea, the Red Sea, the Aden

5    Gulf and the Persian Gulf, with $\delta^{18}O_{oce}$ values reaching $+3.9$ ‰ in this latter. The regional net freshwater export explains, again, these enrichments in $\delta^{18}O_{oce}$ values. Contrary to Werner et al. (2016) who observed enriched $\delta^{18}O_{oce}$ values in the Black





Sea, we obtain depleted $\delta^{18}O_{oce}$ values between $-1$ and $-2.7$ ‰ in this small-scale area, which is in better agreement with the observations. This opposite result is due to the land-sea mask of higher-resolution applied in our model (T63GR15 against T31GR30 used in Werner et al. (2016)) that results in a narrower connection between the Black Sea and the Mediterranean Sea via the Aegean Sea. At high latitudes, the $\delta^{18}O_{oce}$ values are more depleted than the average. In the Southern Ocean, the

modeled $\delta^{18}O_{oce}$ values are between $-0.4$ and $-1$ ‰, in agreement with the observations. The most depleted $\delta^{18}O_{oce}$ values are in the Arctic Ocean, that decrease down to $-13$ ‰. This depletion is mainly caused by the Arctic rivers discharges highly depleted in $\delta^{18}O$ combined with a strong stratification of the simulated Arctic Ocean water masses (not shown).

In a similar way as for the atmospheric part, we compare our simulated $\delta^{18}O_{oce}$ values with the isotopic observations between 0 and 10 m depth (GISS database, see Section 2.3) for a more quantitative evaluation of our model results (Fig. 2b). For the

Atlantic Ocean (blue circles), the Pacific Ocean (red circles), the Indian Ocean (green circles) and the Southern Ocean (brown circles), the model-data comparison does not show any systematic deviations between the modeled $\delta^{18}O_{oce}$ values and the GISS data, characterized by RMSE values lower than 1 ‰ (Atlantic: $r^2 = 0.83$, RMSE $= 0.98$ ‰; Pacific: $r^2 = 0.67$, RMSE $= 0.68$ ‰; Indian: $r^2 = 0.48$, RMSE $= 0.44$ ‰ and Southern: $r^2 = 0.68$, RMSE $= 0.32$ ‰). As already reported in Werner et al. (2016), the deviations from the observations are due to the overestimation of $\delta^{18}O_{oce}$ values near river estuaries around

the Amazon and the Sea of Okhotsk. For the Arctic Ocean, our simulated $\delta^{18}O_{oce}$ are in better agreement with the observations ($r^2 = 0.57$, RMSE $= 1.61$ ‰) compared to the results from Werner et al. (2016) ($r^2 = 0.33$, RMSE $= 2.25$ ‰). However, our simulated values are still more depleted than the observations for many data entries. Because of the strong stratification of the simulated Arctic Ocean water masses, the highly depleted water inflows from Arctic rivers remain in the upper layers of the Arctic Ocean without being well mixed with deeper waters.

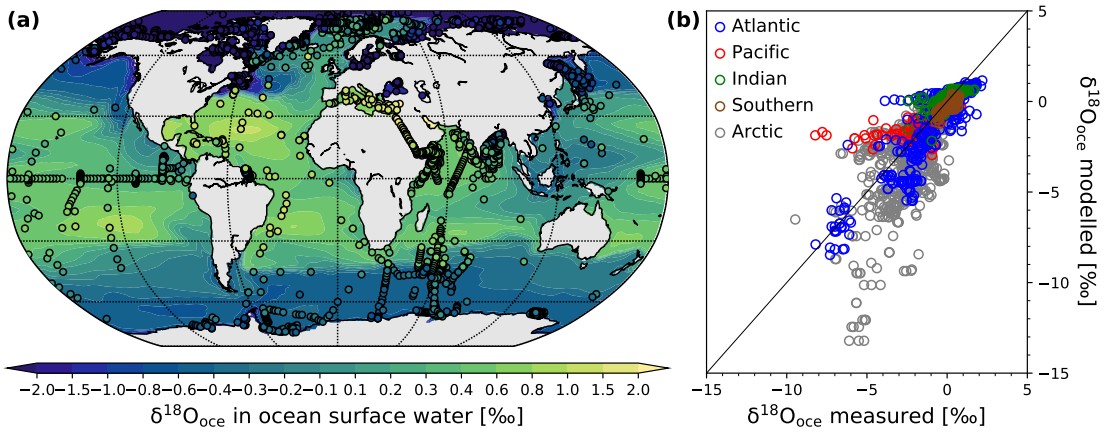

**Figure 2.** (a) Comparison of the global distribution of simulated (background pattern) annual mean $\delta^{18}O_{oce}$ values in ocean surface water (mean over the first 10 meters depth) under PI conditions with observed $\delta^{18}O_{oce}$ values of the GISS database (colored dots). (b) Scatter plot of observed vs. modelled $\delta^{18}O_{oce}$ values for the Atlantic (blue circles), Pacific (red circles), Indian (green circles), Southern (brown circles) and Arctic Oceans (grey circles). The month of sampling has been considered for this scatter plot.



In Fig. 3, we analyze the relationship between $\delta^{18}O_{oce}$ in ocean surface water and salinity for the Atlantic (Fig. 3a), Pacific (Fig. 3b), Indian (Fig. 3c), Southern (Fig. 3d) and Arctic Oceans (Fig. 3e). MPI-ESM-wiso is in fairly good agreement with the observations in terms of absolute values and of $\delta^{18}O_{oce}$–salinity gradients, the latter varying between 0.27 and 0.56 ‰.psu$^{-1}$. The best agreements with the observations are in the Indian and the Pacific Oceans, even if the model is not able to reach the lowest $\delta^{18}O_{oce}$ and salinity values around the Sea of Okhotsk and the Bering Sea. Except for the Pacific Ocean, the relationship between $\delta^{18}O_{oce}$ and salinity in the different basins, expressed by the correlation coefficients $r^2$, is stronger in the model (0.87, 0.90, 0.86, 0.67 and 0.87 for the Atlantic, Pacific, Indian, Southern and Arctic Oceans respectively) than in the observations (0.56, 0.93, 0.70, 0.63 and 0.53 respectively). The main disagreement in the deduced $\delta^{18}O_{oce}$–salinity gradient is in the Atlantic Ocean, with a steeper gradient in MPI-ESM-wiso than in the GISS data. This latter is due to the underestimation by the model of the $\delta^{18}O_{oce}$ values in the North-West Atlantic along the Canadian coast (Fig. 2). Depleted water inflows from Canadian rivers and the strong ocean dynamics of this area with important inter-connected currents, probably not well constrained by MPI-ESM-wiso, can explain the model-data mismatch. In the Arctic and Southern Oceans, even if the modeled $\delta^{18}O_{oce}$–salinity gradient is similar to the observed one, the model underestimates both the $\delta^{18}O_{oce}$ and salinity values probably because of the major roles played by river discharges and changes of sea ice in these areas.

### 3.1.3 Deuterium-excess

The modeling of deuterium-excess signal is challenging for GCMs. For the North Atlantic and Arctic ocean region, the spatial structure of the marine boundary layer water vapor isotopic composition, which greatly influences the dex signal in precipitation, seems to be poorly simulated by the models (Steen-Larsen et al., 2017). Model deficits might be linked to sublimation and moisture source processes over sea ice-covered areas (Bonne et al., 2019; Klein and Welker, 2016). Moreover, in higher latitudes, the representation of dex is very sensitive to supersaturation in polar clouds that is a poorly constrained empirical parameter (Jouzel and Merlivat, 1984; Risi et al., 2013). A comparison of our simulated dex signals with available data represents thus a good evaluation test for our model. Fig. 4 shows the simulated deuterium-excess in precipitation (dex$_p$) and ocean surface water (dex$_{oce}$). The modelled values of dex$_p$ (Fig. 4a) range between 0 and 20 ‰. The highest values are in the northern part of the Sahara and in a $25°$ N – $45°$ N band going from Saudi Arabia to the Himalaya. Lowest values happen in dry regions: the southern Sahara between the latitudes $25°$ N and $10°$ N, Oman and Rajasthan. Low modelled dex$_p$ values (between 2 and 6 ‰) can also be observed over the Southern Ocean, which is an area with large net freshwater input (P−E). For the Antarctic continent, the contrast between the low values of dex$_p$ in the West and high values in the East is well captured by the model, in agreement with the observations. The quantitative model-data comparison (Fig. 4b) shows that the modelled dex$_p$ values are in fairly good agreement with the observations. However, MPI-ESM-wiso tends to underestimate the dex$_p$ values, especially where the observations are higher than 8 ‰.

The modeled dex$_{oce}$ values (Fig. 4c) range between −8 (Persian Gulf) and +7 ‰ (Baltic Sea). We can distinct the mid-to-low latitudinal region of the Atlantic Ocean with lower dex$_{oce}$ values (between −0.2 and −0.8 ‰), the Arctic Ocean where modeled dex$_{oce}$ values vary from 0 ‰ in the North of the Atlantic Ocean to +7 ‰ along the Northern coast of Siberia, and the remaining ocean surface waters with smoother variations in their dex$_{oce}$ composition (from −0.2 to 0.6 ‰). The negative dex$_{oce}$ signal

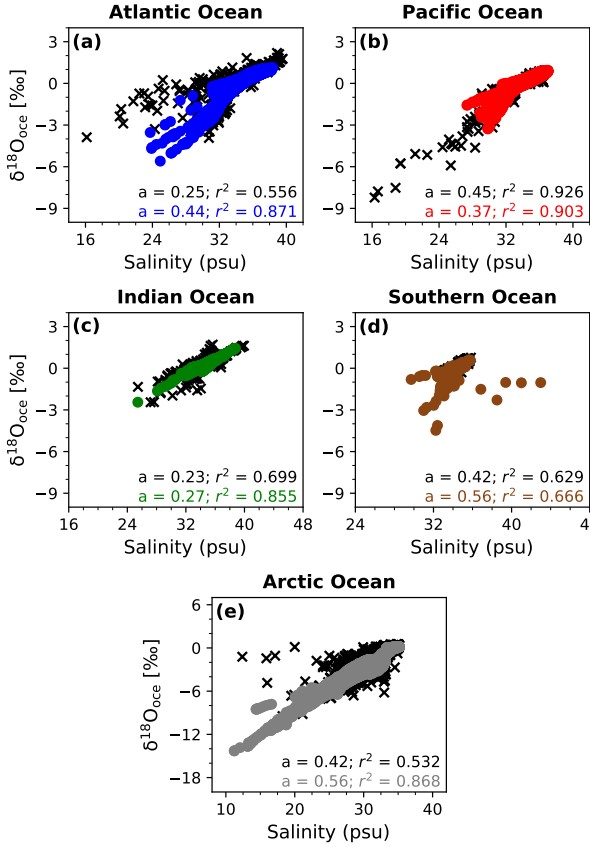

**Figure 3.** Scatter plots of $\delta^{18}O$ in ocean surface water vs. surface salinity for the (a) Atlantic, (b) Pacific, (c) Indian, (d) Southern and (e) Arctic Oceans under PI conditions. The black crosses correspond to the data from GISS database and the colored dots to the modeled values. The gradients and the correlation coefficients of the linear regression fits are given in the legends.

in the mid-to-low latitudinal Atlantic Ocean are due the presence of a net freshwater export. As the exported water masses and the evaporation have a positive deuterium-excess composition, the $dex_{oce}$ in the remaining water becomes more negative due to the hydrological balance. This is the opposite for the areas with positive $dex_{oce}$ values, like in the Baltic Sea and the Arctic Ocean, where there is a surplus of precipitation (positive P−E) with positive deuterium-excess values. The quantitative

5  comparison with the GISS database (Fig. 4d) shows that MPI-ESM-wiso globally overestimates the deuterium-excess values in ocean surface water. Especially, the model is not able to reach the very low values observed in the Mediterranean Sea and overestimates the $dex_{oce}$ values in the Baltic Sea. Moreover, the strong small-scale variations in $dex_{oce}$ observed in the southern Indian Ocean cannot be reproduced by the model because of the too coarse horizontal model resolution.

MPI-ESM-wiso overestimates the deuterium-excess in ocean surface on one side and underestimates the deuterium-excess

10  in precipitation, especially for highly enriched observed values, on the other side. However, the modelled linear relationship between the deuterium-excess in water vapor above the ocean surface ($dex_{vap}$) and the near-surface relative humidity (RH,

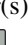



expressed between 0 and 1) is $\mathrm{dex_{vap}} = 50.12 - 52.81 \times \mathrm{RH}$, in very good agreement with the equation given by Pfahl and Sodemann (2014). One possible explanation for the positive and negative biases of modeled dex concentrations in the ocean surface water and the precipitation, respectively, could be due to the used description of fractionation processes during the evaporation of ocean surface water from Merlivat and Jouzel (1979) that would distribute too much dex in ocean surface water

5 and not enough in the water vapor (and so in the precipitation). This agrees with the studies from Steen-Larsen et al. (2014a, b, 2015, 2017) and Bonne et al. (2019), that show biases in the simulated deuterium-excess signal in water vapor in Greenland, North Atlantic and Arctic Ocean in several GCMs compared to in-situ measurements of surface water vapor isotopes.

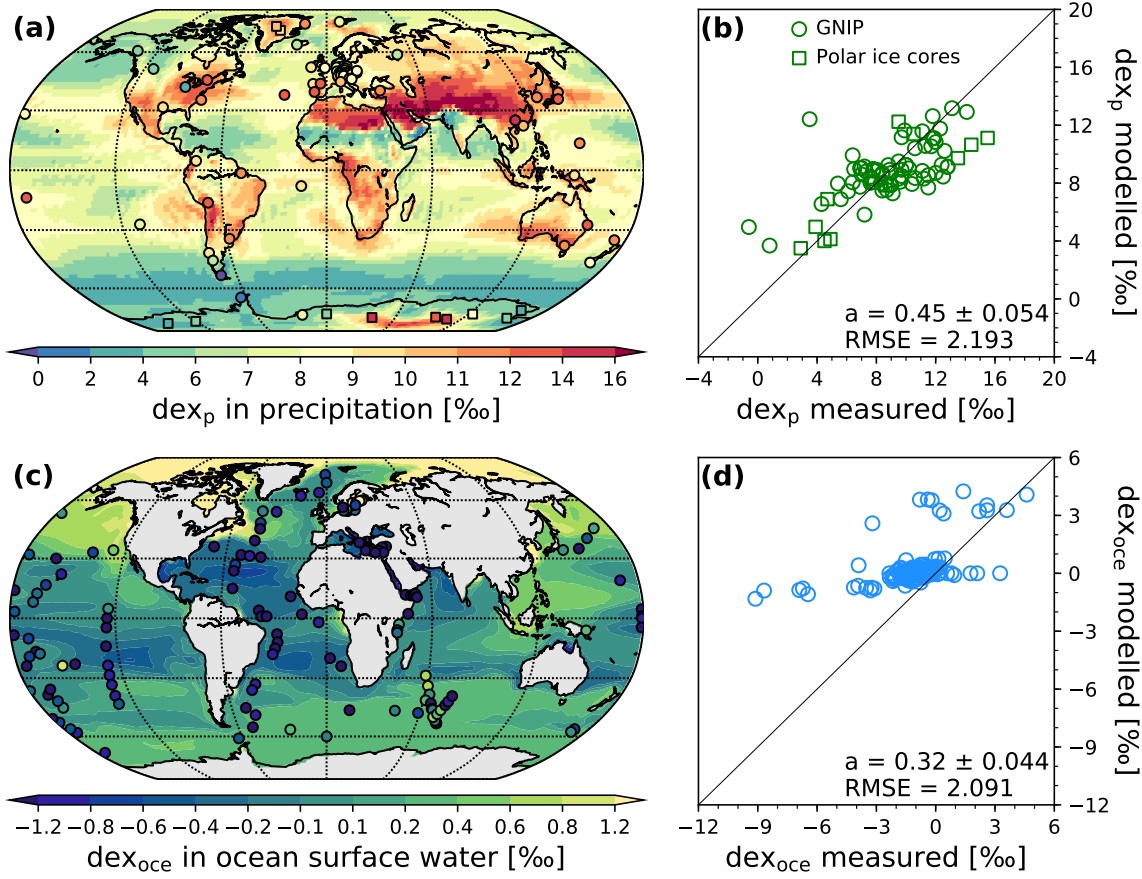

**Figure 4.** Global distribution of simulated and observed annual mean dex values in precipitation (a) and ocean surface waters (c). Scatter plots of modeled vs. observed annual mean deuterium excess values in precipitation (b) and ocean surface waters (d) are shown. The gradients and RMSE of the linear regression fits are given in the legend.





### 3.2 Mid-Holocene simulation

#### 3.2.1 Changes in near-surface air temperature and precipitation

Before analyzing the 6k isotopic anomalies, we check that our modeled 6k−PI anomalies in standard climate variables like the 2m-temperature and the precipitation rate are consistent with previous studies. For that, we show in Fig. 5 the simulated annual

mean, boreal mean winter (DJF) and summer (JJA) changes in 2-meter temperature and precipitation rate between the 6k and PI climates. Because of the different values in orbital parameters and greenhouse gases, the mid-Holocene is characterized by a slightly colder mean global climate compared to the modelled PI ($-0.42°$C). The simulated anomalies in annual mean temperature are rather small, with 6k−PI changes of less than $1°$C (Fig. 5a) in most regions. The exception is the Saharan area where a cooling in the range of $-1$ to $-4°$C can be observed, due to the enhanced African monsoon. We also observe

a slight increase of annual mean temperature over the western Arctic area, northern Siberia, and eastern Europe. As a first result, we conclude that the 6k−PI anomalies in annual mean temperature from MPI-ESM-wiso are globally consistent with the PMIP2 and CMIP5/PMIP3 model results (Harrison et al., 2014). The annual mean change in precipitation amount is very small (less than $1$ mm.year$^{-1}$), in agreement with the previous PMIP2 and CMIP5/PMIP3 model results (Harrison et al., 2014). The African ([$20°$ W – $30°$ E; $10°$ N – $20°$ N] region) and Indian ([$70°$ E – $100°$ E; $20°$ N – $40°$ N] region) monsoons

are enhanced during mid-Holocene by $+1.06$ and $+0.37$ mm.day$^{-1}$ respectively (Fig. 5d), consistent with the PMIP3 results (https://pmip3.lsce.ipsl.fr). The changes in orbital forcing lead to a northward expansion of the African monsoon. This is also in agreement with previous coupled model results (Braconnot et al., 2007; Harrison et al., 2014), even if this monsoon extension is still not large enough compared to the observations (Perez-Sanz et al., 2014).

One of the characteristics of the 6k climate is the enhanced seasonal contrast in the Northern Hemisphere due to changes in

the insolation, giving rise to warmer Northern Hemisphere summers (Fig. 5c). There is a strong land-ocean contrast, with the main positive temperature anomalies on the lands. They range between $+0.5$ and $+3°$C, the highest values being over Northern America and Mongolia, while the 6k−PI summer temperature anomalies in mid and high latitudes over the ocean and the Arctic are lower than $0.5°$C, except near the Greenland coasts. In lower Northern Hemisphere latitudes, the summer surface temperature anomalies over ocean are generally lower (between $0$ and $-1°$C). In the Southern Hemisphere, positive mean JJA

temperature anomalies (i.e. austral winter) can be observed over South America, Australia and coastal west Antarctica. The mean 6k JJA temperature anomalies over the ocean are globally lower, except for some locations in the Southern Ocean near the Antarctic coast. All these results are consistent with the previous PMIP simulations (Braconnot et al., 2007; Harrison et al., 2014). The colder 6k boreal summer in the region from West Africa to India is the consequence of increased precipitation linked to the monsoon changes (Fig. 5f) over this area (Braconnot et al., 2007). The positive anomalies in precipitation over

Africa and India are the strongest during the boreal summer with mean values of $+2.42$ and $+1.00$ mm.day$^{-1}$, respectively. We also find a dipole in the response of the monsoons over the Pacific-Indian area, with enhanced rainfall in the equator sector of the Indian Ocean and reduced rainfall over the Indonesian region. For the mean DJF temperatures (Fig. 5b), we find negative anomalies over the Antarctic continent and positive anomalies over the surrounding Southern Ocean (between $0$ and $0.5°$C). Over the rest of the globe, the 6k−PI anomalies in mean DJF surface temperatures are generally negative.





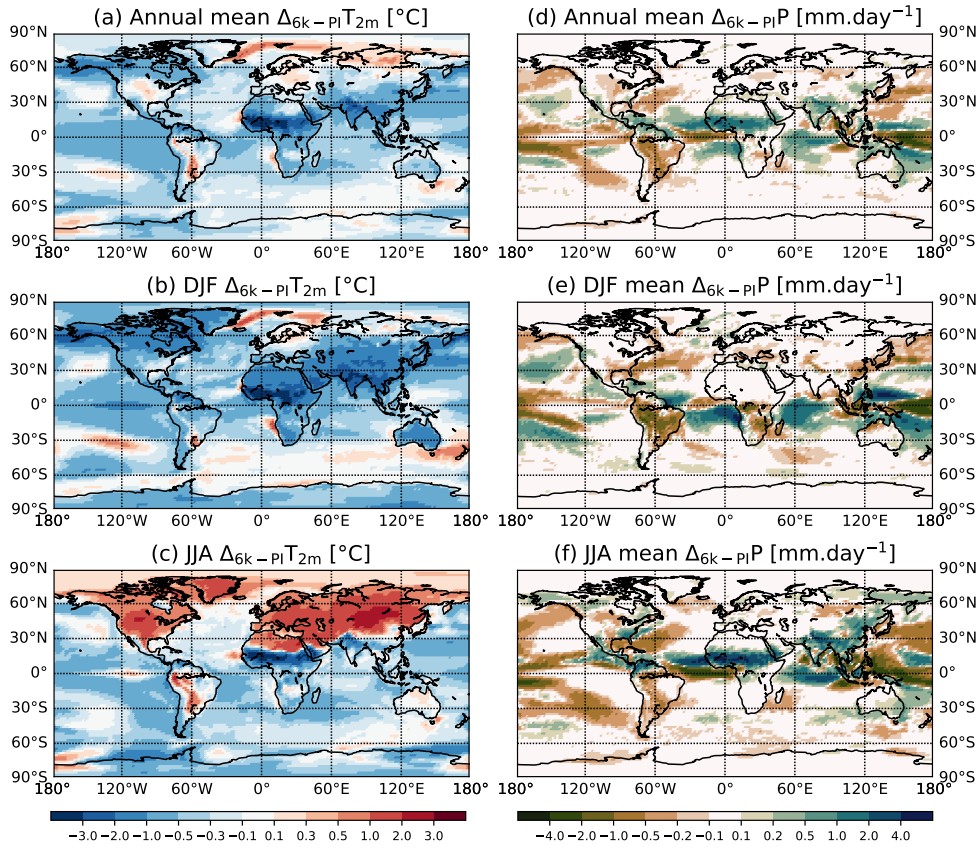

**Figure 5.** Simulated annual, boreal winter (DJF) and boreal summer (JJA) changes in 2m-temperature (a, b, c) and precipitation (d, e, f) between 6k and PI.

### 3.2.2 6k changes in $\delta^{18}$O signals

Even if the changes in temperature and precipitation amount are modest compared to periods like the LGM, they leave imprints on $\delta^{18}$O$_p$ in precipitation values (Fig. 6a). MPI-ESM-wiso simulates a precipitation-weighted mean global decrease in $\delta^{18}$O$_p$ by -0.16 ‰, which is in good agreement with the model results from LeGrande and Schmidt (2009). Positive simulated 6k−PI
5  $\delta^{18}$O$_p$ changes, ranging from +0.3 to +1 ‰, are found over the Arctic area including Greenland, Alaska and the northern part of Siberia. They are likely associated with higher summer temperatures and reductions in Arctic sea ice during 6k (LeGrande and Schmidt, 2009). For the distribution of $\delta^{18}$O$_p$ anomalies over Antarctica, three areas can be distinguished: one region from the 180$^{th}$ meridian to 90° W with anomalies slightly negative or close to zero, another area from 90° W to 100° E with positive anomalies of $\delta^{18}$O$_p$, and the remaining region between 100° E and 180° E with negative isotopic anomalies. Except for
10  Australia, the Indonesian area and some coastal regions in the American continent, negative 6k−PI changes occur in general over the remaining land surfaces. The strongest negative anomalies ($-5.43$ ‰) are located over the Southern Sahara where





strong decrease in surface temperature and amplified African monsoon are simulated by MPI-ESM-wiso. Strong negative changes in $\delta^{18}O_p$ also occur over the Tibetan plateau, with values ranging from $-0.5$ to $-3.5$ ‰. This is probably due to the lower simulated values of annual mean temperature in this area during the 6k period combined with enhanced precipitation rate, especially in summer (Fig. 5). Finally, MPI-ESM-wiso simulates positive 6k−PI anomalies of $\delta^{18}O_p$ between $+0.2$ and

$+1$ ‰ in the West Pacific Ocean and over the Indonesian area, linked to lower rainfall during mid-Holocene.

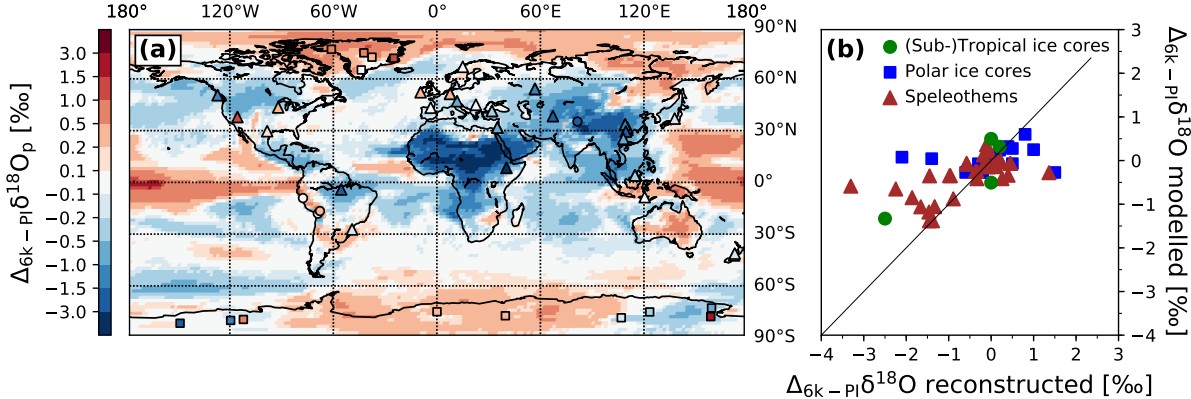

**Figure 6.** (a) Simulated global pattern of annual mean $\delta^{18}O_p$ changes in precipitation between the mid-Holocene and PI climate and comparison with reconstructed $\delta^{18}O$ changes in polar (squares) and (sub-)tropical (dots) ice cores and in calcite speleothems (triangles). (b) Reconstructed $\delta^{18}O$ changes from ice cores and speleothems vs. simulated 6k–PI $\delta^{18}O$ anomalies at the same location. The observed $\delta^{18}O$ anomalies in polar and (sub-)tropical ice core records (blue squares and green dots respectively) are compared to the simulated 6k–PI $\delta^{18}O_p$ changes in precipitation. The observed $\Delta_{6k-PI}\delta^{18}O$ in speleothems (brown triangles) are compared to simulated $\delta^{18}O_c$ changes in calcite (see text).

Next, we compare our simulated 6k−PI $\delta^{18}O_p$ anomalies with isotopic observations from ice cores and speleothems records described in Section 2.3 (Fig. 6b). In general, our modeled isotopic anomalies are in fair agreement with the data ($r^2 = 0.38$ and RMSE $= 0.79$ ‰). The modeled positive $\delta^{18}O_p$ anomalies over most parts of Greenland are confirmed by the polar ice core measurements, as well as the negative anomalies over the southern Greenland coast. The largest deviation is found for the

coastal Renland ice core ($\Delta_{6k-PI}\delta^{18}O_p = +1$ ‰) where MPI-ESM-wiso simulates a too low $\delta^{18}O_p$ anomaly of $+0.25$ ‰. The modeled positive/negative contrast in the $\Delta_{6k-PI}\delta^{18}O_p$ distribution between the central and eastern parts of Antarctica is also found in the data (EDC, Vostok and Talos Dome ice cores on the east; Dome Fuji and EDML on the central area). However, there is a disagreement on the west of the continent with modeled $\delta^{18}O_p$ anomalies close to zero while the observations are clearly positive (WDC ice core) or negative (Byrd and Siple ice cores). At the most eastern region of Antarctica ($160°$ E),

near the Ross Sea, MPI-ESM-wiso is not able to catch the positive 6k−PI $\delta^{18}O_p$ anomaly at the Taylor Dome site. Concerning the $\delta^{18}O$ anomalies from calcite in speleothems, a majority of our simulated 6k−PI isotopic changes are of the same sign than the speleothem data (22 on 33 records, including an uncertainty of $\pm0.05$ ‰ as Comas-Bru et al. (under review)). The model reproduces well the observed negative and positive 6k−PI changes in $\delta^{18}O_c$ over the Tibetan plateau and the coastal





areas of the South American continent, respectively. Disagreements with the speleothem data are found in the US and in Europe where observed positive anomalies are not captured by MPI-ESM-wiso. The largest deviations are found for two speleothems located in Ethiopia ($\Delta_{6k-PI}\delta^{18}O_{model} = -0.59\,\text{‰}$ and $\Delta_{6k-PI}\delta^{18}O_{reconstructed} = -3.31\,\text{‰}$) and in the Great Basin of western North America ($\Delta_{6k-PI}\delta^{18}O_{model} = -0.28\,\text{‰}$ and $\Delta_{6k-PI}\delta^{18}O_{reconstructed} = +1.36\,\text{‰}$). These discrepancies likely

reflect an insufficient amplification of precipitation rate (or its wrong location) over eastern Africa and a too weak increase of temperature over Northeast America during the mid-Holocene period. More generally, the amplitude of the modeled $\delta^{18}O$ changes at speleothem sites is underestimated by MPI-ESM-wiso. This is likely related to the underestimation by the model of the 6k changes in climate variables like temperature and precipitation rate, as already noticed in previous model studies (Harrison et al., 2014).

Concerning the changes in $dex_p$ in precipitation, we find negative anomalies over Antarctica and Greenland. The modeled $dex_p$ value at EDC site is of opposite sign compared to the measured value ($\Delta_{6k-PI}dex_{model} = -0.45\,\text{‰}$ and $\Delta_{6k-PI}dex_{obs} = +0.7\,\text{‰}$) while the Greenland values are consistent with the observations (GRIP: $\Delta_{6k-PI}dex_{model} = -0.28\,\text{‰}$ and $\Delta_{6k-PI}dex_{obs} = -0.2\,\text{‰}$; NGRIP: $\Delta_{6k-PI}dex_{model} = -0.20\,\text{‰}$ and $\Delta_{6k-PI}dex_{obs} = -0.5\,\text{‰}$).

Fig. 7 shows our modeled annual mean changes in $\delta^{18}O_{oce}$ in ocean surface water between 6k and PI. The simulated annual

mean $\delta^{18}O_{oce}$ change between 6k and PI is very small ($-0.01\,\text{‰}$), in agreement with previous model results (LeGrande and Schmidt, 2009). The model simulates an enrichment of $\delta^{18}O_{oce}$ in the Arctic Ocean ranging from $0.1$ to $0.7\,\text{‰}$, except around the $180^{th}$ meridian, which is related to Arctic sea ice reductions and warmer summers in 6k relative to PI. This modeled isotopic enrichment is observed in the precipitation, too. Enhanced runoff with more depleted $\delta^{18}O$ values (not shown) are associated with negative 6k−PI anomalies in $\delta^{18}O_{oce}$ along the coasts of central America and South-Western Africa, Red Sea,

Persian Gulf, and in the Bay of Bengal. As for the changes in $\delta^{18}O_p$, MPI-ESM-wiso simulates a dipole of enriched/depleted $\delta^{18}O_{oce}$ values in the West Pacific Ocean (from $+0.05$ to $+0.5\,\text{‰}$) and the Bay of Bengal (from $-0.05$ to $-1\,\text{‰}$), respectively. Positive $\delta^{18}O_{oce}$ changes are also found in the sub-tropical latitudes of the East Pacific Ocean. The enrichment of $\delta^{18}O_{oce}$ during 6k relative to PI is due to the lower annual mean precipitation rates over these areas and vice versa. Slight positive $\delta^{18}O_{oce}$ anomalies are also simulated along the western Antarctic coast, related to the higher 6k summer temperatures over the

Southern Ocean.

## 4   Temporal relationships between the water isotopes and climate variables

The classical use of water isotopes to reconstruct the past variations of climate implies that the modern spatial relationship between isotopes and climate variables, such as surface temperature, precipitation rate or salinity, can be taken as a surrogate for the temporal isotope–climate gradient at a given site. Such temporal relationships can be calculated from our model results.

In Section 4.1, we first look at the interannual variability between water isotopes and 2m-temperature, precipitation rate and salinity under PI conditions. We limit the analysis to the interannual gradients from our PI simulation as they are qualitatively similar to the ones derived from our 6k simulation. Then, we examine the long-term temporal 6k−PI $\delta^{18}O$ gradients versus the different climate variables in the Section 4.2.





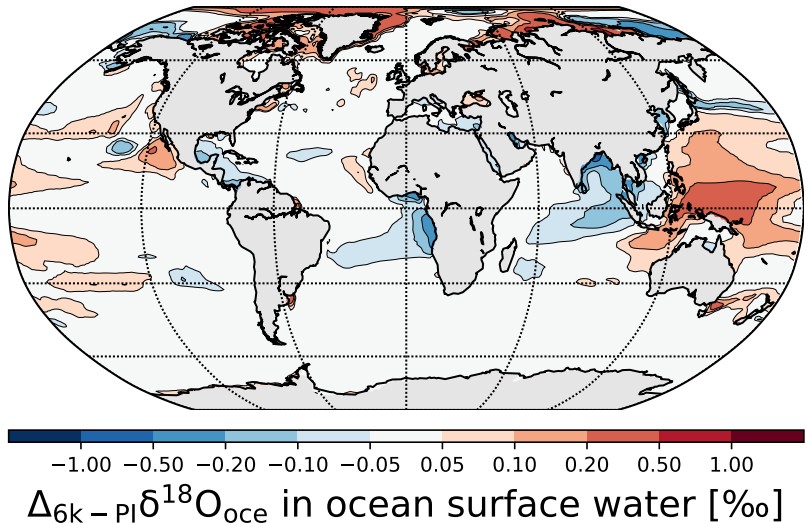

$$\Delta_{6k-PI}\delta^{18}O_{oce} \text{ in ocean surface water [‰]}$$

**Figure 7.** Modeled annual mean $\delta^{18}O_{oce}$ changes in ocean surface water between the mid-Holocene and PI climate.

## 4.1 Interannual relationships of water isotopes and climate variables for the PI climate

In the same way as previous studies (Schmidt et al., 2007; Risi et al., 2010b; Roche and Caley, 2013), we calculate for each grid box the interannual relationship (correlation coefficients and gradients) between monthly anomalies of $\delta^{18}O_p$ and temperature and precipitation rate over the 150 years of our PI simulation. These monthly anomalies are calculated by subtracting from each

monthly mean value the multi-year mean value of the corresponding month, e.g. we subtracted the long-term January mean value from the January values. For the following, we consider only the grid boxes with a temporal correlation higher than 0.4 in absolute value (Risi et al., 2010b; Roche and Caley, 2013). By introducing such a correlation threshold, we remove any grid cell with a negative $\delta^{18}O_p$–temperature gradient (where precipitation rates are higher) and/or positive $\delta^{18}O_p$–precipitation gradient (where precipitation rates are too low) that are not physically plausible. The temporal correlations of $\delta^{18}O_p$ to temperature

are positive (Fig. 8a) in the mid- to high-latitude grid boxes, i.e. over Antarctica, Northern America, Greenland, Europe and northern part of Russia. At these locations, the interannual $\delta^{18}O_p$–temperature gradients vary between 0.3 and 0.9 $‰.°C^{-1}$, with the highest values over the poles (Fig. 8b). For Greenland, we find a mean PI interannual $\delta^{18}O_p$–temperature gradient of 0.57 $‰.°C^{-1}$, averaged over all Greenland ice core locations (Table 1) where the correlation coefficient is higher than 0.4 in absolute value. This is less than our modelled PI spatial gradient of 0.71 $‰.°C^{-1}$ (calculated by considering the 25 grid cells

centered on each drill location, excluding the ocean grid points), in agreement with previous modeling studies (Werner et al., 2000; Schmidt et al., 2007). For the ice core locations in East and West Antarctica, the averaged modern spatial gradients are of 0.76 and 0.88 $‰.°C^{-1}$ respectively, in agreement with the mean observed value of 0.8 $‰.°C^{-1}$ (Masson-Delmotte et al., 2008) and previous model results (Schmidt et al., 2007; Werner et al., 2018). A clear distinction can be made between East and West when looking at the PI interannual $\delta^{18}O_p$–temperature gradients. For East Antarctica, we obtain a value of 0.66 $‰.°C^{-1}$,



close to the modern spatial gradient over this area. On the contrary, the mean interannual gradient at West Antarctic ice core sites is of $0.39\,\%o.°C^{-1}$ only, which is more than 2 times smaller than the corresponding average spatial gradient. This result, which could be related to sea ice variations or large-scale transport of moisture from the ocean, will be investigated in a further study by the comparison with other models results.

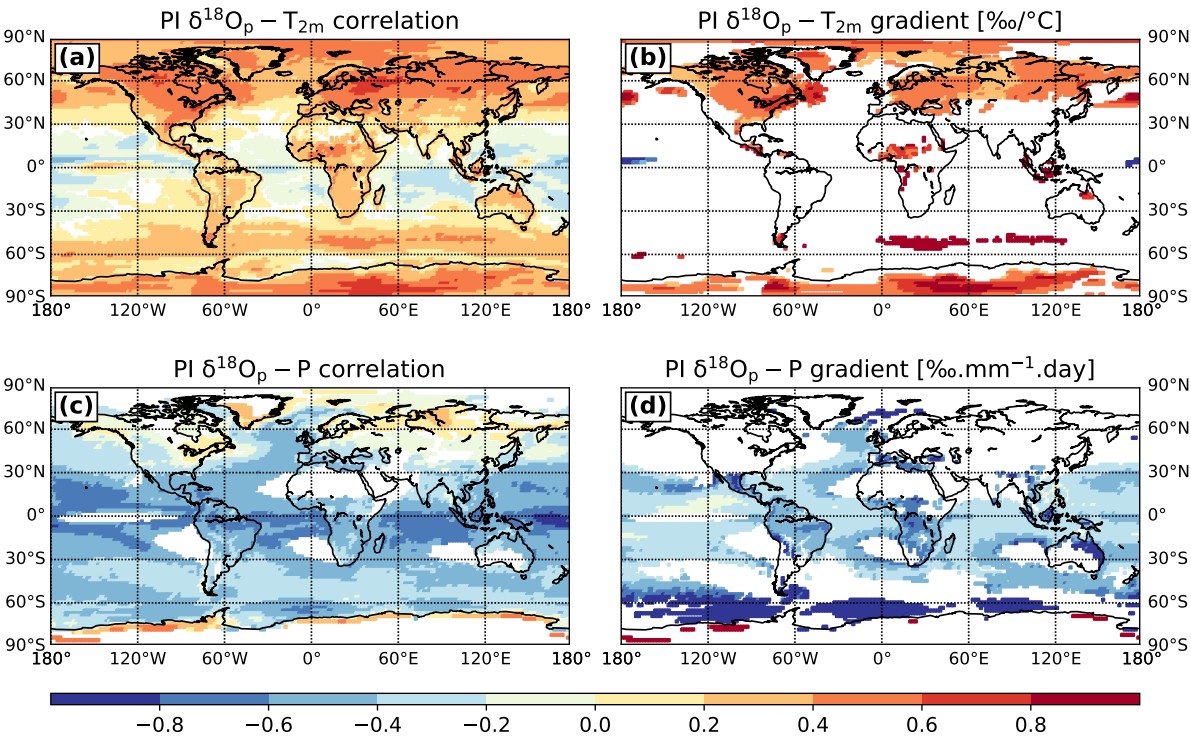

**Figure 8.** Correlation coefficients (a, c) and gradients (b, d) of the interannual relationship between monthly anomalies of $\delta^{18}O_p$ in precipitation and temperature (a, b) and precipitation rate (c, d). All shown values are significant at 95 % level. We restrict the analysis of the $\delta^{18}O_p$–precipitation relationship at grid points with mean precipitation rate higher than $250\,\mathrm{mm.year}^{-1}$. The gradient values are only shown for correlation coefficients higher than 0.4 in absolute value.

5   The correlation values between modeled monthly anomalies of $\delta^{18}O_p$ and precipitation rate are negative from the equator to the mid-latitudes (Fig. 8c). This area gathers the Indian, Pacific and Atlantic Oceans, the Central and South America, and a part of the African continent. The interannual $\delta^{18}O_p$–precipitation gradients vary from $-0.2$ to $-0.8\,\%o.\mathrm{mm}^{-1}.\mathrm{day}$ (Fig. 8d). The lowest values are located over the Amazonian area, the Central America and in the south of Sahara, with temporal gradients steeper than $-0.5\,\%o.\mathrm{mm}^{-1}.\mathrm{day}$. For example, we find a mean value of $-0.61\,\%o.\mathrm{mm}^{-1}.\mathrm{day}$ over the African monsoon area,

10  consistent with previous model results (Schmidt et al., 2007; Risi et al., 2010b). We obtain an interannual $\delta^{18}O_p$–precipitation mean gradient of $-0.38\,\%o.\mathrm{mm}^{-1}.\mathrm{day}$ at cells where the annual mean temperature is equal or greater than $+20°C$, consistent with the observed and modeled spatial PI gradients ($-0.46$ and $-0.36\,\%o.\mathrm{mm}^{-1}.\mathrm{day}$ respectively).





In a similar way to the atmospheric relationships, we assess the temporal gradients between monthly anomalies of $\delta^{18}O_{oce}$ in ocean surface water and salinity under PI conditions. The modeled $\delta^{18}O_{oce}$ PI interannual variations are strongly correlated to the salinity changes (Fig. 9a) almost everywhere, with a mean correlation coefficient $r = 0.82$ (standard deviation $\sigma = 0.19$). Lowest correlation values ($r$ between 0.2 and 0.8) are located at the high latitudes near the Antarctic coasts and in the Arctic area, due to the presence of sea ice, and in several (sub-)tropical areas (west of Sahara and [170° E – 100° W; 10° N – 30° S] region) probably because of the influence of precipitation amounts on the isotopic concentrations. The mean value of the PI temporal gradients is of 0.33 ‰.psu$^{-1}$ (Fig. 9b). Generally, gradients are steeper in the mid- to high-latitudes (between 0.3 and 0.7 ‰.psu$^{-1}$) and shallower in the tropics (between 0.1 and 0.3 ‰.psu$^{-1}$), in agreement with previous model results (Schmidt et al., 2007; LeGrande and Schmidt, 2011). In the West Pacific/Indian Ocean area, the gradients are slightly steeper than in the rest of the tropical ocean (more than 0.3 ‰.psu$^{-1}$), possibly due to that region's central role in exporting water vapor to the extra-tropics (Schmidt et al., 2007). We obtain mean PI interannual $\delta^{18}O_{oce}$–salinity gradients of 0.30, 0.29, 0.29, 0.38 and 0.40 ‰.psu$^{-1}$ for the Atlantic, Pacific, Indian, Southern and Arctic Oceans respectively. Except for the Indian Ocean, the mean modeled spatial gradients (Fig. 3) are steeper than the interannual ones, in agreement with previous model results (Holloway et al., 2016).

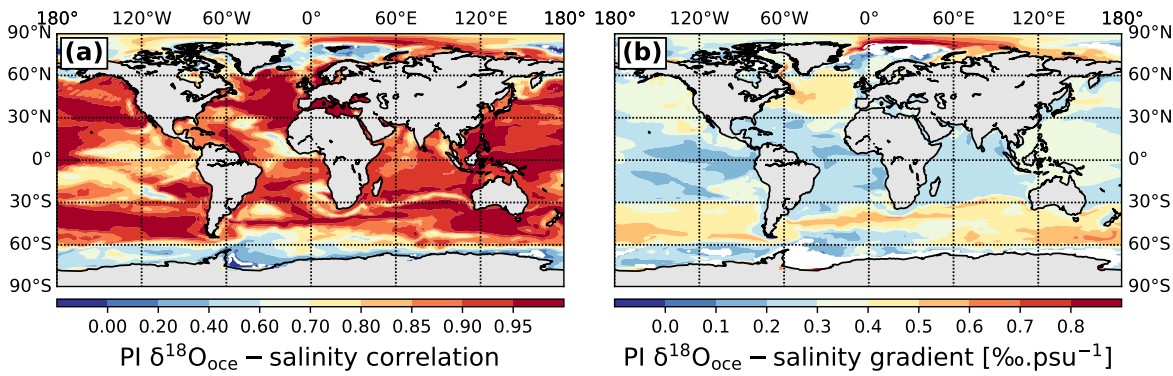

**Figure 9.** Correlation coefficients (a) and gradients (b) of the interannual relationship between monthly anomalies of $\delta^{18}O_{oce}$ in ocean surface water and salinity. All shown values are significant at 95 % level. The gradient values are only shown for correlation coefficients higher than 0.4.

## 4.2 Temporal relationships of water isotopes and climate variables between 6k and PI

The global spatial relationship between climate variables and water isotopes does not change significantly between our simulated mean climate states of 6k and PI. For example, we find a mean spatial $\delta^{18}O_p$–temperature regression gradient of $0.63 \pm 0.014$ ‰.°C$^{-1}$ for the 6k simulation, similar to the PI one (Fig. 1c). The surface relationships between $\delta^{18}O_{oce}$ and salinity in the different oceans for the PI (Fig. 3) and 6k periods are extremely similar, too. Now, we examine the calculated 6k−PI temporal relationship between different climate variables (temperature, precipitation and ocean salinity) and $\delta^{18}O$





changes. If we take as an example the temporal relationship with temperature ($T$) changes, the temporal gradient $m$ of each grid cell is calculated as $m = (\delta^{18}O_{p,6k} - \delta^{18}O_{p,PI})/(T_{6k} - T_{PI})$. For this latter, we restrict our calculation to the grid boxes where the simulated mean temperature is below $+20°C$ for both PI and 6k. Moreover, we select only the grid cells showing an absolute temperature 6k−PI difference of at least $0.5°C$. We present the results in a histogram (Fig. 10a) and global maps (Fig.

10b, c and d). By using the values of mean annual temperatures (MAT) and $\delta^{18}O_p$, the calculated temporal 6k−PI gradient is below the spatial PI gradient (dotted line in Fig. 10a) in 79 % of the grid cells considered (red bars in Fig. 10a). Only 10.1 % of the selected grid cells (229 on 2273) have a temporal gradient between 0.5 and 0.7 ‰.$°C^{-1}$, close to the simulated PI spatial gradient. Upon examination of the corresponding map (Fig. 10b), it appears that the $\delta^{18}O_p$–$T$ temporal relationship can have negative (north of the Canada, Alaska, western coast of South America) or very high gradients (over China and north of India)

at certain locations. For the first case, the small difference between the modeled 6k and PI annual mean temperatures (less than $1°C$, Fig. 5a) and a strong seasonality of precipitation can probably lead to meaningless $\delta^{18}O_p$–$T$ gradients (Gierz et al., 2017). For the second case, changes in the monsoon strength can explain the very high $\delta^{18}O_p$–$T$ gradients (more precipitation combined with lower temperatures). One can also notice that only a very few of grid cells over Greenland and Antarctica, where the correlation between the isotopic content in precipitation and the temperature is high, are in accordance with the

selection criteria, described above.

For numerical reasons, robust $\delta^{18}O_p$–$T$ gradients can only be calculated for non-negligible 6k−PI temperature anomalies. In a next step, we therefore analyze not the mean annual temperature values but the modeled mean values of the warmest month, i.e. the mean temperature values of July for the Northern Hemisphere and of January for the Southern Hemisphere (MTWA: mean temperature of the warmest month). The frequency distribution of the temporal gradients using this new calculation

corresponds to the green bars in Fig. 10a (MTWA$_{0.5}$), and the global map is shown in Fig. 10c. From this map, we can see that $\delta^{18}O_p$–$T$ temporal gradients can be calculated for the grid boxes at mid to high latitudes. This is reflected by an higher proportion of 6k−PI temporal gradient values that are between 0.5 and 0.7 ‰.$°C^{-1}$ (13.1 % of the grid cells, i.e. 583 of 4438). The resulting mean 6k−PI temporal gradients around ice core locations in East Antarctica, West Antarctica and Greenland are 0.52, 0.52 and 1.36 ‰.$°C^{-1}$ respectively. These temporal gradients calculated from our 6k and PI isotopic simulations

are substantially different from the modeled surface gradients: higher for Greenland and lower for East and West Antarctica. Because a non-negligible portion of the analyzed grid cells still reveals a negative $\delta^{18}O_p$–$T$ temporal gradient, we increase the temperature cutoff from 0.5 to $1°C$ (WTMA$_1$). Despite this stronger restriction, the proportion of the grid boxes having $\delta^{18}O_p$–$T$ gradients values between 0.5 and 0.7 ‰.$°C^{-1}$ remains almost the same (11.8 % of the grid cells, i.e. 226 of 1908). However, applying this temperature restriction removes many of the grid boxes with negative or very high gradients (blue bars

in Fig. 10a), like in central Antarctica (Fig. 10d). The mean 6k−PI temporal gradients around Greenland and East Antarctic ice core locations are equal to 0.77 and 0.82 ‰.$°C^{-1}$, respectively. These values are very close to the modeled spatial PI gradients of 0.71 and 0.77 ‰.$°C^{-1}$. These results could mean that the spatial gradients are more a surrogate of summer temperature variations between 6k and PI, especially over Greenland where the seasonality is enhanced in 6k compared to PI. This result has to be taken in caution, especially for the East Antarctic area where only the EDC site satisfies the required conditions for





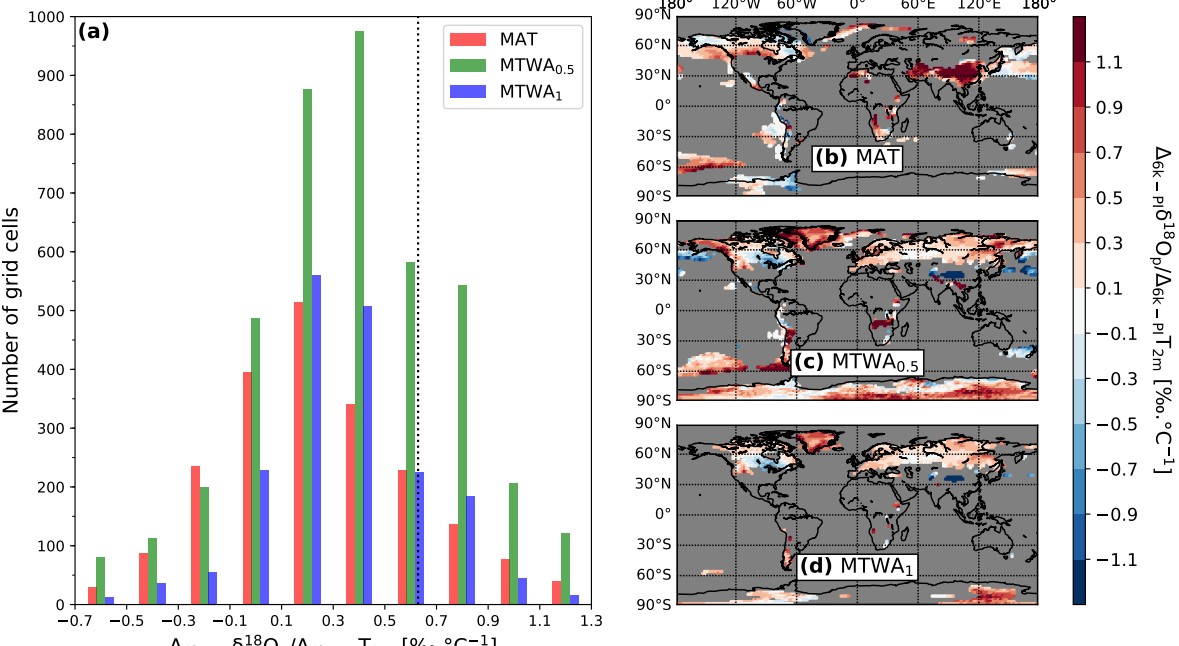

**Figure 10.** (a) Histogram of the calculated temporal 6k–PI $\delta^{18}O_p$–$T$ gradients for all grid cells where (i) simulated mean temperature for both PI and 6k is lower than $20°C$ and (ii) absolute change in temperature between the 6k and PI control simulations is at least $0.5°C$ (MAT, WTMA$_{0.5}$) or $1°C$ (WTMA$_1$). The dotted line shows the simulated mean spatial PI $\delta^{18}O_p$–$T$ gradient of $0.63$ ‰.$°C^{-1}$. The gradients are calculated in three different ways: with the annual mean $\delta^{18}O_p$ and temperature values where $|\Delta_{6k-PI}MAT| \geq 0.5°C$ (MAT, red bars), with the mean $\delta^{18}O_p$ and temperature values of the warmest month where $|\Delta_{6k-PI}WTMA| \geq 0.5°C$ (WTMA$_{0.5}$, green bars), with the mean $\delta^{18}O_p$ and temperature values of the warmest month where $|\Delta_{6k-PI}WTMA| \geq 1°C$ (WTMA$_1$, blue bars). Their spatial distribution is shown in (b), (c) and (d) respectively.

the gradient calculation in the East Antarctic area. Under this new condition, it is also not possible to calculate a mean temporal $\delta^{18}O_p$–$T$ gradient for West Antarctic ice core locations.

The $\delta^{18}O_p$ values in polar regions grid boxes might be biased by strong changes of seasonality or intermittency of the precipitation rate (Sime et al., 2009). So, in the same way as Gierz et al. (2017), we replace the arithmetic annual mean

5   temperatures by the precipitation-weighted annual mean temperatures in the calculation of the $\delta^{18}O_p$–$T$ gradients (Fig. 11). With this new calculation, we obtain mean $6k-PI$ temporal gradients of $0.58$, $0.38$ and $0.01$ ‰.$°C^{-1}$ for the grid cells around the Greenland, East Antarctic and West Antarctic ice core locations, respectively. This shows well the effects of seasonality on the $\delta^{18}O_p$–$T$ gradients over Greenland and, in a less extent, over East Antarctica. The great variability of the resulting mean temporal gradients for the West Antarctic ice core locations (e.g., mean values near to zero, positive or not meeting the

10   requirements for temperature difference) could indicate that temperature changes between the warm 6k and PI periods are not the main driver of the variations of $\delta^{18}O_p$ in this area. The contrast in the $\delta^{18}O_p$–$T$ gradient between West Antarctica, that is





more sensitive to moisture inputs from coast regions, and East Antarctica, the stronger isolated plateau region, is also visible in the $\delta^{18}O_p$ anomalies between 6k and PI (Fig. 6a).

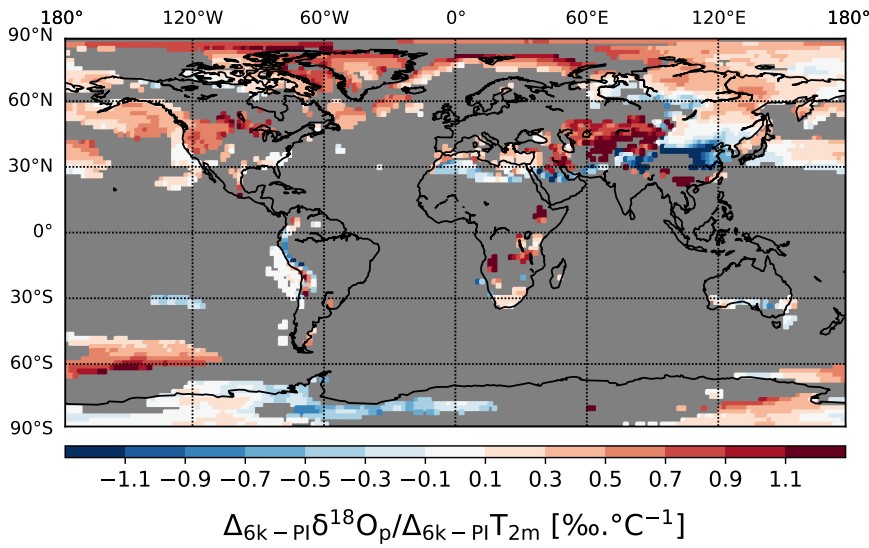

$$\Delta_{6k-PI}\delta^{18}O_p / \Delta_{6k-PI}T_{2m} \ [\text{‰.}°C^{-1}]$$

**Figure 11.** Spatial distribution of the calculated temporal 6k–PI $\delta^{18}O_p$–$T$ gradients for all grid cells where (i) simulated annual mean temperature for both PI and 6k is lower than $20°C$ and (ii) absolute change in temperature between the 6k and PI control simulations is at least $0.5°C$. The gradients are calculated in the same way as in the Fig. 10b but with the use of precipitation-weighted temperatures instead of the arithmetic annual mean temperatures.

The $\delta^{18}O_p$ in precipitation over West Antarctica could be sensitive to a more important contribution of relatively enriched evaporating water from the surrounding ocean near the coast during 6k, in link with the increased divergence of sea ice and

5 enhanced open water areas around the coast (Noone and Simmonds, 2004). We also observe slight positive austral summer (DJF) sea surface temperature anomalies (between 0 and $0.5°C$) in the western Southern Ocean area, while the entire Antarctic continent is cooler (middle left map of Fig. 5), and an increase between 5 and 20 % of the evaporation flux around the Antarctic coasts. This sea ice hypothesis can be checked by looking at the changes in the vertically integrated water vapor transport over Antarctica between PI and 6k climates (Fig. 12). We focus on the warmest season (DJF) and see that the western part of the

10 continent is clearly exposed to water vapor input coming from regions near the Antarctic Peninsula, the Ross Sea and the Amundsen Sea. But no significant differences in the water vapor transport pattern can be observed between our PI and 6k climates. However, positive annual mean 6k−PI anomalies of $\delta^{18}O$ in vertically integrated water vapor (between 0.1 and 0.4 ‰) are simulated by the model over the Southern Ocean, which cannot be simply explained by changes in temperature as these latter are not strong enough. Thus, insolation variations apparently lead to seasonality changes that alter the $\delta^{18}O_p$ in

precipitation signal independent of temperature changes over the polar regions during the mid-Holocene.

According to our model results, the Indian and African monsoons are enhanced during the mid-Holocene compared to the pre-industrial period (Section 3.2.1). As a consequence, the precipitation over these areas is more depleted in heavy water





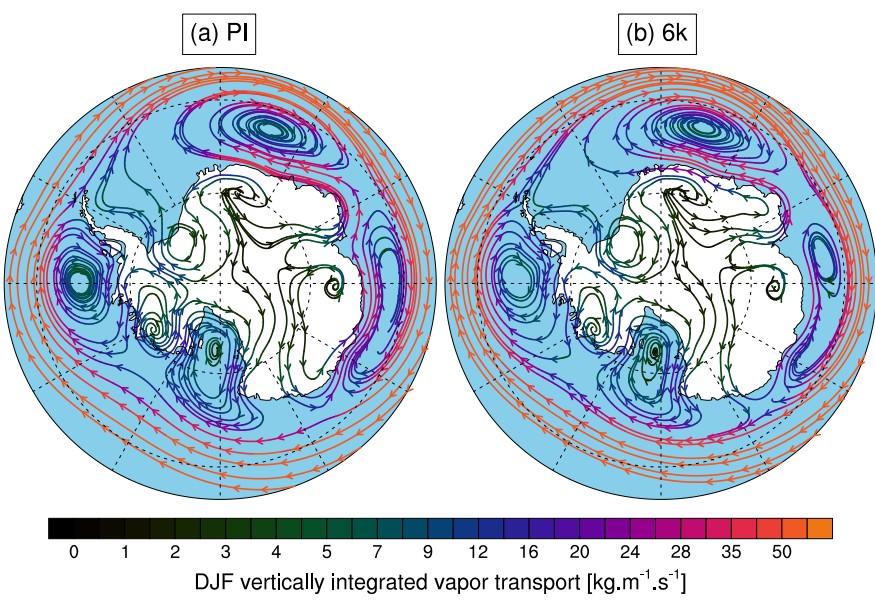

**Figure 12.** Vertically integrated water vapor transport over Antarctica during austral summer for PI (a) and 6k (b).

isotopes, consistent with the amount effect (Fig. 6). The modeled mean 6k−PI temporal $\delta^{18}O_p$–precipitation gradient over the African monsoon for the summer months (JJA) is equal to $-1.52$ ‰.mm$^{-1}$.day, which is higher in absolute value than the interannual gradient (Section 4.1). If we consider annual averages instead of JJA values, the mean $\delta^{18}O_p$–precipitation gradient is even steeper with a value of $-4.15$ ‰.mm$^{-1}$.day. Our results indicate that the amount effect may strongly depend

on the considered time scale and that the use of a calibration based on a present-day interannual scale can be misleading for reconstructing past precipitation rates (Schmidt et al., 2007; Risi et al., 2010b). This is also indicated by the very high regional variability of the 6k−PI $\delta^{18}O_p$–precipitation gradients over the African monsoon area (standard deviation of 1.90 ‰.mm$^{-1}$.day for the mean JJA $\delta^{18}O_p$–precipitation gradient).

Fig. 13 presents a distribution map of the temporal 6k−PI $\delta^{18}O_{oce}$–salinity gradients for every oceanic surface grid cell. The

calculation is restricted to the grid cells where the 6k−PI absolute change in salinity is equal or higher than a threshold value of 0.02 psu. We find a global mean gradient of 0.36 ‰.psu$^{-1}$, close of the modeled interannual mean gradient. However, the mean 6k−PI $\delta^{18}O_{oce}$–salinity gradients in the different oceans can be significantly different from the mean modeled interannual gradients in these same oceans. For the Atlantic Ocean, the averaged 6k−PI $\delta^{18}O_{oce}$–salinity gradient is of 0.22 ‰.psu$^{-1}$, which is shallower than the modeled interannual gradient (Fig. 9b), itself shallower than the spatial PI gradient (Fig. 3a).

However, strong 6k−PI gradients are simulated in the North Atlantic (between 0.6 and 1.25 ‰.psu$^{-1}$), much higher than the PI interannual ones (0.4-0.6 ‰.psu$^{-1}$). The mean gradient in the Pacific Ocean is similar to the interannual gradient, but higher values are modeled in the 10° N – 30° N area, linked to the 6k−PI changes in precipitation rate. The mean temporal 6k−PI $\delta^{18}O_{oce}$–salinity gradient in the Indian Ocean (0.37 ‰.psu$^{-1}$) is higher than the corresponding mean PI interannual and





spatial gradients. Especially, we notice steep gradient values near the Bay of Bengal (coincident with more depleted $\delta^{18}O_{oce}$ values) due to the enhanced runoff during the mid-Holocene period (Section 3.2.2). The average of the 6k−PI gradients in the Southern Ocean is similar to the interannual one ($0.37\,\text{‰.psu}^{-1}$) while the mean gradient in the Arctic Ocean is similar to the $\delta^{18}O_{oce}$–salinity spatial relationship ($0.56\,\text{‰.psu}^{-1}$). However, there is a strong spatial variability in the calculated gradients

because of the changes in sea ice coverage and/or weak 6k−PI difference in salinity.

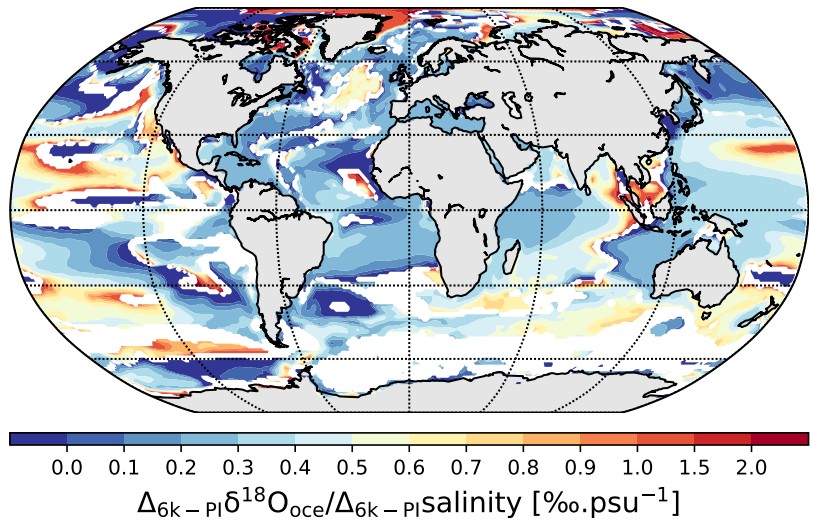

**Figure 13.** Spatial distribution of calculated temporal 6k–PI $\delta^{18}O_{oce}$–salinity gradients for oceanic grid cells where the 6k–PI absolute change in salinity is at least $0.02\,\text{psu}$.

We conclude that the reconstruction of past salinity through isotopic content in sea surface waters can be complicated for regions with strong ocean dynamics (North Atlantic Ocean), variations in sea ice regimes (Arctic and Southern Oceans) or significant changes in freshwater budget (Bay of Bengal), giving an extremely variable relationship between $\delta^{18}O_{oce}$ and salinity over small spatial scales. This reconstruction task is even more difficult because of the small 6k−PI changes in salinity

that can lead to large uncertainties in the calculated $\delta^{18}O_{oce}$–salinity gradients.

## 5    Conclusions and perspectives

In this study, we present the first simulations of the fully coupled model MPI-ESM, enhanced with water isotope diagnostics. The water isotopes have been implemented in all the components of the model (atmosphere, dynamic vegetation, hydrological discharge, ocean/sea-ice) and the related isotope masses of $H_2{}^{16}O$, $H_2{}^{18}O$ and $HD^{16}O$ are fully exchanged between the atmo-

sphere and the ocean. The model has been run successfully for 2500 model years under PI and 6k conditions, each. The PI spatial distribution of modeled isotopes in precipitation and ocean surface water has been evaluated against present-day observations from the GNIP and GISS database, ice cores and speleothems. For precipitation, we find a good to very good agreement



of $\delta^{18}O_p$ values with the observational data. Especially, the modeling of $\delta^{18}O_p$ over Antarctica is improved compared to the previous model release ECHAM5/MPIOM (Werner et al., 2016) through the better ability of our model to reach the lowest temperatures, due to overall model enhancements and a higher spatial resolution. Our modeled $\delta^{18}O_{oce}$ in ocean surface water is in fairly good agreement with the isotopic observations from the GISS database. For the Atlantic, Pacific and Indian Oceans,

the main model-data deviations are found near the river estuaries, where the coarse resolution of MPIOM hampers a realistic simulation of water mass mixing near the coastal regions. In the Arctic ocean, improvements in the $\delta^{18}O_{oce}$ model-data agreement are found compared to Werner et al. (2016) but many model values are still too depleted compared to the observations. This could be related to the influence of sea ice in the area and/or to the inadequate mixing of highly depleted water from Arctic rivers into the ocean. The PI simulated values of the second-order parameter d-excess are in fairly good agreement with

the isotopic observations in precipitation and ocean surface water. MPI-ESM-wiso underestimates the deuterium-excess values in precipitation and, at the same time, overestimates the deuterium-excess values in ocean surface water. This pattern, already observed by Werner et al. (2016) with ECHAM5/MPIOM, suggests that the approach by Merlivat and Jouzel (1979) used in our model setup to describe the fractionation processes during the evaporation over the ocean should maybe revised in the future. Finally, the simulated modern spatial relationships between isotopes and climate variables (2m-temperature, precipitation rate

and salinity) are in good agreement with the observed ones.

     The modeled changes in temperature and precipitation rate during the mid-Holocene compared to the pre-industrial period are consistent with previous PMIP results, with a warmer northern hemisphere summer and enhanced African and Indian monsoons. One great advantage of enabling MPI-ESM to model water stable isotopes is the possibility to directly compare available isotopic measurements with our climate simulations. We find a fair agreement between our modeled 6k isotopic

anomalies and the observations from ice cores and speleothems. MPI-ESM-wiso simulates higher $\delta^{18}O_p$ values over Greenland linked to higher mid-Holocene summer temperatures and changes in sea ice. Over Antarctica, $\delta^{18}O_p$ anomalies reveal three different regions of change: no isotope changes over the west (180° W – 90° W), positive $\delta^{18}O_p$ anomalies over the center (90° W – 100° E) and negative anomalies over the most eastern part of the continent (100° E – 180° E). Over the East Antarctic plateau, the negative anomalies are consistent with the ice core measurements and are likely related to variations in local

temperature. The modeled positive 6k−PI changes in $\delta^{18}O_p$ over the [90° W – 100° E] Antarctic area, more influenced by evaporating waters from the Southern Ocean, are in agreement with the observations, too. The absence of $\delta^{18}O_p$ anomalies, according to MPI-ESM-wiso, in the western part of the Antarctic continent, disagrees with the available observations. In the tropics, $\delta^{18}O_p$ and $\delta^{18}O_{oce}$ variations are linked to changes in precipitation rate (amount effect), i.e. enhanced African and Indian monsoons with more depleted $\delta^{18}O$ values, and lower precipitation rate with enriched $\delta^{18}O$ values over the tropical

West Pacific Ocean and the Indonesian area.

     In numerous previous paleoclimate studies, one of the main assumptions for using water isotopes to study past climate variations is that the modern spatial isotope/climate variable relationships can be used as a surrogate for the temporal gradients at any specific site. In this study, we focused especially on the variability of these relationships during and between two distinct periods of the Holocene. For that, we have analyzed the modeled temporal isotope–climate gradients (i) at an interannual time

scale in our PI simulation and (ii) between the mean 6k and PI climates. For the $\delta^{18}O_p$–temperature relationship, we find that





the interannual gradients over Greenland and East Antarctica are slightly lower than the corresponding modern spatial gradients, in agreement with previous studies (Schmidt et al., 2007; Risi et al., 2010b). Concerning the 6k−PI temporal gradients for these same areas, they are very close to the gradients retrieved from the spatial relationships. However, it should be noted that the temporal gradients for 6k−PI changes were analyzed by using the mean temperatures of the warmest month because of the

very small difference in simulated annual mean temperatures between the 6k and PI periods. Our result could highlight an effect of strong seasonality, but it needs confirmation with the use of other models. For West Antarctica, we find a rather low PI interannual gradient (more than 2 times lower than the modern spatial one) and no gradient for 6k−PI changes because of the rather weak temperature changes over this region. Moreover, the close to zero gradient calculated by using precipitation-weighted mean temperature values indicates that the seasonality is not one of the drivers of $\delta^{18}O_p$ changes in this region. Our results

indicate that mid-Holocene changes in $\delta^{18}O_p$ over West Antarctica, an area more sensitive to water vapor changes over nearby coastal ocean regions, are not mainly controlled by local temperature variations. Concerning the link between the water isotope variations and the changes in precipitation rate over the (sub-)tropics, we find that the spatial and PI interannual relationships are similar. For 6k−PI changes, the amount effect is stronger and depends on the considered period (JJA or annual mean), in agreement with previous modeling studies (Schmidt et al., 2007; Risi et al., 2010b). Our model results reveal that it can be

difficult to reconstruct past variations in precipitation amount for different climatic conditions (enhanced African monsoons for example) based on the modern isotope-precipitation relationship. Finally, the spatial relationships between surface salinity and $\delta^{18}O_{oce}$ in the different oceans are in general higher than the interannual PI gradients. The reconstruction of surface salinity for the mid-Holocene climate can be subject to errors because of the large regional variability of the 6k−PI $\delta^{18}O_{oce}$–salinity gradient, due to different factors like ocean dynamics, sea ice changes or changes in the freshwater budget.

The focus of this study on the mid-Holocene and pre-industrial climates was a first step for studying the isotope-climate relationship under different warm climate conditions by MPI-ESM-wiso model simulations. Future studies will investigate the hydrological cycle variability for other interglacial periods, including the LIG, and for a transient Holocene experiment.

*Code availability.* MPI-ESM is available under the Software License Agreement version 2 after acceptance of a license (https://www.mpimet.mpg.de/en/science/models/license/). The isotopic version MPI-ESM-wiso is available upon request on the AWI's FusionForge repos-

itory (https://swrepo1.awi.de/projects/mpi-esm-wiso/).

*Author contributions.* AC developed the model code, designed the experiments and performed the simulations with the help of MW. AC and MW analyzed the model outputs. AC wrote the manuscript with contributions from all co-authors.

*Competing interests.* The authors declare that they have no conflict of interest.



*Acknowledgements.* This work was supported by the German Federal Ministry of Education and Research (BMBF) as a Research for Sustainability initiative (FONA) through PalMod project (FKZ: 01LP1511B). All simulations were performed at the German Climate Computing Center (DKRZ). We thank Christian Stepanek for his help on experiments design and the fruitful discussions. We acknowledge Dirk Barbi for his help for debugging, installing and running the model.



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
