# Peer review of "Water isotopes – climate relationships for the mid-Holocene and pre-industrial period simulated with an isotope-enabled version of MPI-ESM"

_Climate of the Past, 2019_

## Referee Comment (RC1) · Jonathan Holmes (Referee) · 16 Jul 2019

cp-2019-72 Water isotopes – climate relationships for the mid-Holocene and pre-industrial period simulated with an isotope-enabled version of MPI-ESM Author(s): Alexandre Cauquoin et al.

General comments This manuscript describes the enhanced isotope-enabled version of the MPI-ESM Earth-system model. Climate models associated with isotope diagnostics are becoming more common. They are an increasingly important component

of the palaeoclimatological 'toolkit'. Given prevalence of isotopic proxies within palaeo-climate archives and the difficulties that are often associated with converting palaeoclimate proxies, including those based on stable isotopes, into estimates of temperature and precipitation, it makes sense to equip the models with isotope diagnostics rather than attempt to cover the proxies into more 'traditional' estimates of palaeoclimate. Comparison of different isotope-enabled models is well established through the Stable Water Isotope Intercomparison Group (Phase 2 - https://data.giss.nasa.gov/swing2/). Comparisons of isotope records from marine, ice core and terrestrial archives with output from isotope-enabled models have been completed for the present and past (see, for example, Sturm et al., 2010; Werner, 2010; Jones and Dee, 2018). Detailed descriptions and performance evaluations for new or enhanced models are important, hence this MS is well suited to CoP. The paper is well structured and generally well argued and written. I have a few specific comments on the content as well as some minor suggestions for improvement in the language, which are detailed below.

The authors begin with a well-reasoned account of the rationale behind isotope-enabled models. They then describe the isotope-enabled MPI-ESM model in some detail, along with the results of simulations for the pre-industrial and mid Holocene (=6ka) intervals and the modern-day and paleo-water isotope datasets used for model evaluation. They finally examine pre-industrial – 6ka differences in the data and in the model and compare spatial and temporal gradients in the atmosphere and oceans, which have particular relevance to the interpretation of paleo-isotope records.

I have general familiarity with isotope-enabled models, but do not have the technical expertise to be able to comment in detail on the model setup and simulations: I focus instead on the data-model comparisons.

Specific comments Page 2, line 30-32. You could also cite Pfahl and Sodemann (2014), which you cite elsewhere, and also Fröhlich et al. (2002), which additionally lists moisture recycling and evaporation of falling raindrops as controls on the deuterium excess. Also, isn't d a more usually symbol for the deuterium excess?

Page 3 line 35 – page 4 line 4. You could add reference to the freshwater hosing experiments in HadCM3 (Tindall and Valdes, 2011) and the comparison of the results of those experiments with palaoe-isotope data from lake sediments (Holmes et al., 2016).

Section 2.3 Observation data. How representative of pre-industrial conditions are the observation data? The ocean water and GNIP data are certainly not pre-industrial: the speleothem data span the pre-industrial and the post-industrial period. None of the datasets could are exclusively pre-industrial. While this may not be a problem, the authors should at least discuss the mismatch and any implications.

Page 13 Line 24 – rephrase, as it appears that low values are found both in dry and in humid regions if I interpret your results correctly.

Page 13 Line 24 – Rajasthan (India)

Page 14 Line 19 – Not clear what you mean by 'on one side'

Page 26 Line 3 – Isn't this quite surprising given that most rainfall occurs in summer in such regions?

Page 27 Line 1-2 – isn't there a similar pattern, but not as well expressed, in the Arabian Sea?

References Holmes, J. A., Tindall, J., Roberts, N., Marshall, W., Marshall, J. D. Bingham, A., Feeser, I., O'Connell, M., Atkinson, T., Jourdan, A-L., March, A., Fisher, E. H. (2016) Lake isotope records of the 8200-year cooling event in western Ireland: Comparison with model simulations. Quaternary Science Reviews, 131, 341-349.

Jones, M.D. and Dee, S.G., 2018. Global-scale proxy system modelling of oxygen isotopes in lacustrine carbonates: New insights from isotope-enabled-model proxy-data comparison Quaternary Science Reviews. 202, 19-29

Froehlich, K., Gibson, J.J., Aggarwal, P.K., 2002 Deuterium excess in precipitation

and its climatological significance. International conference on study of environmental change using isotope techniques; Vienna (Austria); 23-27 Apr 2001, p.54-66.

Pfahl, S., Sodemann, H., 2014. What controls deuterium excess in global precipitation? Climate of the Past, 10, 771-781.

Sturm, C., Zhang, Q., Noone, D., 2010. An introduction to stable water isotopes in climate models: benefits of forward proxy modelling for paleoclimatology. Climate of the Past, 6, 115-129.

Tindall, J.C., Valdes, P.J., 2011. Modeling the 8.2 ka event using a coupled atmosphere-ocean GCM. Glob. Planet. Change 79, 312-321.

Werner, M. 2010 Modelling stable water isotopes: Status and perspectives EPJ Web of Conferences 9, 73–82

Technical corrections General The authors make common use of phrases that would undoubtedly disturb isotope 'purists': examples include 'depletion in isotopic composition' (p1, line 19), 'depleted isotopic values' (p1, line 22 and elsewhere), 'depletion of delta18Op' (p10, lines 9 and 11, and elsewhere) amongst others. I know that opinion is divided over such terminology and that some authors regards its use as heretical, whereas others regard such authors as puritanical pedants. I leave it to the present authors and editor to decide in this case. Table 2.1 in Chapter 2 of Principles of Stable Isotope Geochemistry, 2nd Edition, by Zachary Sharp (available for free download at https://digitalrepository.unm.edu/unm_oer/1/ provides careful guidance in case the authors wish to follow the purists, or should the editor compel them to do so!

Specific Page 1 Line 12 and passim 'In link with' is a slightly strange phrase – 'linked to' would be better.

Page 4 Line 12 '. . .seasonal changes in insolation. . .' perhaps?

Line 15 Which part? The Monsoon domain? Clarify.

[Figure]

Line 21 'near-surface air temperature' Also 'ocean salinity'

Page 7 Line 28 '. . .are both at 0‰

Page 13 Line 24 'are found' rather than 'happen'

Line 31 'distinguish between' rather than 'distinct the'

Page 18 Line 12 Taylor

Line 14 Siple Dome

Page 19 Line 18 '(not shown) is. . .'

Page 23 Line 21 '. . .a higher'

---

## Referee Comment (RC2) · Anonymous Referee #2 · 6 Sep 2019

In this study, Cauquoin et al. conducted a set of time slice experiments with newer version of isotope-enabled coupled climate model, namely MPI-ESM-wiso, and comprehensively validated the results by fully using the currently available isotopic data over the world. Moreover, they made analyses on how isotopic information can be proxy of climate information by using isotope-temperature, isotope-precipitation, isotope-salinity relationships. In conventional method, isotope-climate relationship is assumed to be stable (meaning that the same linear relationship is assumed for both climates), but it is highly doubtful. This study revealed that such simple relationship is indeed not same

in different climates because the isotope information is determined by complicated processes.

The manuscript is very well written. The results are nicely illustrated by the figures, and the findings and conclusions are logically reasonable and convincing. Thus I have only minor comments.

1. Abstract is perhaps too long. So that the important essence of the paper is diluted. I would like the authors to make the abstract more concise.

2. In abstract and conclusions, the authors cautioned that interpretation of isotope information is more complex than previously thought. It is true, but is there any recommendation?

3. Almost all abbreviations are directly used without telling the long names.

4. Figure 4c and 4d show that the modeled sea water D-excess is significantly less fluctuated than the observation. But isn't it due to the layer thickness? The observed depth is very shallow, so surface kinetic fractionation is highly influential. For more appropriate comparison, some sort of simulator (for bucket sampling?) would be needed.

5. Mid-Holocene climate is shown in 3.2.1, and the authors try to explain its plausibility. But isn't it simply the same as the MPI-ESM results? If so, the part can be omitted only by referring appropriate paper for PMIP6.

6. Figure 8 and 9 show isotope-climate relationships in pre-industrial period. Why don't you show the same quantities for MH and the difference between PI and MH?

---

## Author Comment (AC1) · 30 Sep 2019

We acknowledge Jonathan Holmes and the anonymous referee for their reviews and constructive comments that helped to improve this manuscript. We have revised it as described in detail below, and we hope that we have dealt with all suggestions in an adequate manner. For the corrections, we provide page and line numbers from the revised manuscript with track changes. The references cited can be found in the manuscript.

**Referee 1 (Jonathan Holmes)**

General comments This manuscript describes the enhanced isotope-enabled version of the MPI-ESM Earth-system model. Climate models associated with isotope diagnostics are becoming more common. They are an increasingly important component of the palaeoclimatological 'toolkit'. Given prevalence of isotopic proxies within palaeoclimate archives and the difficulties that are often associated with converting palaeoclimate proxies, including those based on stable isotopes, into estimates of temperature and precipitation, it makes sense to equip the models with isotope diagnostics rather than attempt to cover the proxies into more 'traditional' estimates of palaeoclimate. Comparison of different isotope-enabled models is well established through the Stable Water Isotope Intercomparison Group (Phase 2 - https://data.giss.nasa.gov/swing2/). Comparisons of isotope records from marine, ice core and terrestrial archives with output from isotope-enabled models have been completed for the present and past (see, for example, Sturm et al., 2010; Werner, 2010; Jones and Dee, 2018). Detailed descriptions and performance evaluations for new or enhanced models are important, hence this MS is well suited to CoP. The paper is well structured and generally well argued and written. I have a few specific comments on the content as well as some minor suggestions for improvement in the language, which are detailed below.

The authors begin with a well-reasoned account of the rationale behind isotope-enabled models. They then describe the isotope-enabled MPI-ESM model in some detail, along with the results of simulations for the pre-industrial and mid Holocene (=6ka) intervals and the modern-day and paleo-water isotope datasets used for model evaluation. They finally examine pre-industrial – 6ka differences in the data and in the model and compare spatial and temporal gradients in the atmosphere and oceans, which have particular relevance to the interpretation of paleo-isotope records.

I have general familiarity with isotope-enabled models, but do not have the technical expertise to be able to comment in detail on the model setup and simulations: I focus instead on the data-model comparisons.

Specific comments Page 2, line 30-32. You could also cite Pfahl and Sodemann (2014), which you cite elsewhere, and also Fröhlich et al. (2002), which additionally lists moisture recycling and evaporation of falling raindrops as controls on the deuterium excess. Also, isn't d a more usually symbol for the deuterium excess?

We added these references (p3, lines 5-6). Concerning the symbol for the deuterium excess, it is true that we find in the literature d or d-excess. We choose the second one for more clarity. We also changed the figure 4 accordingly.

Page 3 line 35 – page 4 line 4. You could add reference to the freshwater hosing experiments in HadCM3 (Tindall and Valdes, 2011) and the comparison of the results of those experiments with palaoe-isotope data from lake sediments (Holmes et al., 2016).
We added these references (p4, lines 7-9).

Section 2.3 Observation data. How representative of pre-industrial conditions are the observation data? The ocean water and GNIP data are certainly not pre-industrial: the speleothem data span the pre-industrial and the post-industrial period. None of the datasets could are exclusively pre-industrial. While this may not be a problem, the authors should at least discuss the mismatch and any implications.
It is true that this difference of climate state between the observations and our model results should be discussed. For the ocean, we do not expect big changes between pre- and post-industrial values because of the inertia of the system (p8, lines 32-34). Concerning the atmospheric GNIP data, our modeled temperature values could be lower than in the case of a present-day simulation. Of course, it means that our $\delta^{18}O$ values are maybe lower because of this different climate state. However, it would probably not significantly change the relationship between the water isotopes and the temperature (or precipitation) (p8, lines 23-26). For the SISAL speleothem data, the use of an extended modern baseline (1850– 1990 CE) increases the data uncertainties by only ±0.5 ‰ (Comas-Bru et al., 2019) (p9, lines 13-14). We added these statements in the section 2.3 Observational Data.

Page 13 Line 24 – rephrase, as it appears that low values are found both in dry and in humid regions if I interpret your results correctly.
Ok (p15, lines 1-3): 'Lowest values are found in dry regions like the southern Sahara between the latitudes 25° N and 10° N, Oman and Rajasthan (India) as well as over the Southern Ocean (between 2 and 6 ‰), which is…'

Page 13 Line 24 – Rajasthan (India)
Ok (p15, line 2)

Page 14 Line 19 – Not clear what you mean by 'on one side'
We removed this expression for more clarity (p15, lines 23-24).

Page 26 Line 3 – Isn't this quite surprising given that most rainfall occurs in summer in such regions?
As you can see in the figure 5 of the manuscript, the 6k-PI JJA average anomaly in precipitation over the African monsoon is more than 2 times bigger than the 6k-PI annual mean change in precipitation over the same area. For the $\delta^{18}O_p$, the difference between the JJA and annual mean anomalies is much smaller because these values are precipitation-weighted. It explains why we obtain a steeper mean $\delta^{18}O_p$- precipitation gradient ($\Delta\delta^{18}O_p/\Delta P$) with the annual mean values than with the JJA ones. This result is in agreement with the findings of Risi et al. (2010b). We added this explanation in the manuscript (from p26 line 16 to p27 line1).

Page 27 Line 1-2 – isn't there a similar pattern, but not as well expressed, in the Arabian Sea?

Yes, the runoff is also enhanced during 6k in this area. We added the following sentence in the manuscript (p27, lines 17-18): '… period (Section 3.2.2). This pattern, even if it is not as well expressed, is also visible in the Arabian Sea. The average…'

Technical corrections General The authors make common use of phrases that would undoubtedly disturb isotope 'purists': examples include 'depletion in isotopic composition' (p1, line 19), 'depleted isotopic values' (p1, line 22 and elsewhere), 'depletion of delta18Op' (p10, lines 9 and 11, and elsewhere) amongst others. I know that opinion is divided over such terminology and that some authors regards its use as heretical, whereas others regard such authors as puritanical pedants. I leave it to the present authors and editor to decide in this case. Table 2.1 in Chapter 2 of Principles of Stable Isotope Geochemistry, 2nd Edition, by Zachary Sharp (available for free download at https://digitalrepository.unm.edu/unm_oer/1/ provides careful guidance in case the authors wish to follow the purists, or should the editor compel them to do so!

We did our best to follow these recommendations. Especially, we corrected all along the text the expressions implying the words depleted/depletion and enriched/rich.

Specific Page 1 Line 12 and passim 'In link with' is a slightly strange phrase – 'linked to' would be better.

Ok

Page 4 Line 12 '. . .seasonal changes in insolation. . .' perhaps?

Ok (p4, line 21)

Line 15 Which part? The Monsoon domain? Clarify.

'So, the mid-Holocene is characterized by an enhanced seasonal contrast in the Northern Hemisphere with warmer summers in this part of the Earth, and by a strengthening of the African, Indian and Asian monsoons.' (p4, lines 23-25)

Line 21 'near-surface air temperature' Also 'ocean salinity'

We guess you mean line 31. It's corrected (p5, line 7).

Page 7 Line 28 '...are both at 0‰

Ok (p8, line 3)

Page 13 Line 24 'are found' rather than 'happen'

Ok (p15, line 1)

Line 31 'distinguish between' rather than 'distinct the'

Ok (p15, line 9)

Page 18 Line 12 Taylor

We suppose that you mean Talos instead of Talos Dome, done (p19, line 5).

Line 14 Siple Dome

Ok (p19, line 7)

Page 19 Line 18 '(not shown) is. . .'
Ok (p20, line 14)

Page 23 Line 21 '. . .a higher'
Ok (p25, line 6)

---

## Author Comment (AC2) · 30 Sep 2019

We acknowledge Jonathan Holmes and the anonymous referee for their reviews and constructive comments that helped to improve this manuscript. We have revised it as described in detail below, and we hope that we have dealt with all suggestions in an adequate manner. For the corrections, we provide page and line numbers from the revised manuscript with track changes. The references cited can be found in the manuscript.

**Referee 2 (anonymous)**

In this study, Cauquoin et al. conducted a set of time slice experiments with newer version of isotope-enabled coupled climate model, namely MPI-ESM-wiso, and comprehensively validated the results by fully using the currently available isotopic data over the world. Moreover, they made analyses on how isotopic information can be proxy of climate information by using isotope-temperature, isotope-precipitation, isotope-salinity relationships. In conventional method, isotope-climate relationship is assumed to be stable (meaning that the same linear relationship is assumed for both climates), but it is highly doubtful. This study revealed that such simple relationship is indeed not same in different climates because the isotope information is determined by complicated processes.
The manuscript is very well written. The results are nicely illustrated by the figures, and the findings and conclusions are logically reasonable and convincing. Thus I have only minor comments.

1. Abstract is perhaps too long. So that the important essence of the paper is diluted. I would like the authors to make the abstract more concise.
We made changes in the abstract accordingly.

2. In abstract and conclusions, the authors cautioned that interpretation of isotope information is more complex than previously thought. It is true, but is there any recommendation?
Concerning West Antarctica, the coupling with an ice sheet model or the use of a zoomed grid over this area could help to better describe the role of the water vapor transport and sea ice. A systematic isotope model inter-comparison study for further insights on model-dependency of these results would be beneficial, too (abstract: p2 lines 2-5 and lines 9-10; conclusion: p30 lines 9-13 and lines 21-22).

3. Almost all abbreviations are directly used without telling the long names.
We checked that the long names of the corresponding acronyms are stated in the manuscript (MPI-ESM, SISAL, GISS, IAEA, HadCM3….).

4. Figure 4c and 4d show that the modeled sea water D-excess is significantly less fluctuated than the observation. But isn't it due to the layer thickness? The observed depth is very shallow, so surface kinetic fractionation is highly influential. For more appropriate comparison, some sort of simulator (for bucket sampling?) would be needed.
We added a statement about potential model-data mismatches due to different vertical layer thicknesses, as suggested by the referee (p15, lines 20-21). We also mention in the initial manuscript the too coarse horizontal resolution in MPIOM as a solution for the southern Indian Ocean data-model mismatch. We rephrased this sentence, as the resolution affects the model-data comparison on a global scale (p15, lines 21-22).

5. Mid-Holocene climate is shown in 3.2.1, and the authors try to explain its plausibility. But isn't it simply the same as the MPI-ESM results? If so, the part can be omitted only by referring appropriate paper for PMIP6.

Yes, to add the isotopes in the MPI-ESM does not change the values of the 'standard' variables like temperature. However, according to our knowledge, this is the first time that the 6k results are shown with this recent release of the model (MPI-ESM1.2, see Mauritsen et al., 2019).

6. Figure 8 and 9 show isotope-climate relationships in pre-industrial period. Why don't you show the same quantities for MH and the difference between PI and MH?

As we declared at the beginning of the section 4, the figures 8 and 9 for 6k are similar to the PI ones. For avoiding repetitions, we did not put them in the manuscript. We added the figures in Supplementary Materials (that you can see below) and a statement in the introduction part of the section 4 (p 21, lines 7-8).

[Figure]

Figure S1: As Figure 8 but for 6k.

[Figure]

Figure S2: Distribution of the differences between the 6k $\delta^{18}O_p$ – temperature gradients and the PI ones (a). The same for the $\delta^{18}O_p$ – precipitation gradients (b).

[Figure]

Figure S3: As Figure 9 but for 6k.

[Figure]

Figure S4: Distribution of the differences between the 6k $\delta^{18}O_{oce}$ – salinity gradients and the PI ones.